# REPRESENTATIONAL SIMILARITY VIA INTERPRETABLE VISUAL CONCEPTS

**Neehar Kondapaneni**[1]    **Oisin Mac Aodha**[2]    **Pietro Perona**[1]
[1]Caltech    [2]University of Edinburgh

## ABSTRACT

How do two deep neural networks differ in how they arrive at a decision? Measuring the similarity of deep networks has been a long-standing open question. Most existing methods provide a single number to measure the similarity of two networks at a given layer, but give no insight into *what* makes them similar or dissimilar. We introduce an interpretable representational similarity method (RSVC) to compare two networks. We use RSVC to discover shared and unique visual concepts between two models. We show that some aspects of model differences can be attributed to unique concepts discovered by one model that are not well represented in the other. Finally, we conduct extensive evaluation across different vision model architectures and training protocols to demonstrate its effectiveness. `Project Page: RSVC  Code: github.com/nkondapa/RSVC`

## 1 INTRODUCTION

The accuracy of deep neural networks has steadily increased over the last few years thanks to improvements in model architectures, dataset size, and pretraining strategies. However, much less is understood regarding *how* the representations of different models have changed to make the models more effective. Thus, there is growing interest in developing methods that allow practitioners to compare different networks. Comparing the activation matrices of two neural networks over the same set of inputs underpins current *representational similarity* methods, e.g., CCA (Hotelling, 1936), CKA (Kornblith et al., 2019), RSA (Kriegeskorte et al., 2008), and Brain-score (Schrimpf et al., 2018). While these approaches provide a score denoting the similarity between two different models, they do not identify the specifics of what makes two models' computations similar or dissimilar, and what aspects of a representation lead to differences in model decisions.

In parallel, methods for concept-based eXplainable AI (XAI) have improved our ability to understand what features individual models use to arrive at decisions. Understanding these features is critical for ensuring model fairness and identifying potential sources of bias (Kaminski & Urban, 2021; Kop, 2021). In general, XAI methods sacrifice model fidelity to produce explanations that are simple enough for human interpretation (Fel et al., 2023a; Cunningham et al., 2024). Thus, there is a tension between model fidelity and human understanding.

We propose that contrasting two models is an effective way to identify and highlight what makes a model unique, so that users can identify critical features that drive differences in model behavior. To investigate this idea we develop a new approach that extends concept-based XAI methods so they can quantitatively measure representational similarity and provide interpretable insights into the differences between models. We name our method *Representational Similarity via Interpretable Visual Concepts* (RSVC). Our method builds on interpretability approaches in which the activations of a given layer are decomposed into a coefficient matrix and a vector basis. These approaches group images that produce similar activation patterns together. Thus, we define a visual concept to be the equivalence class of images that produce similar activation patterns. After computing visual concepts from each model, our method evaluates whether the visual concepts from one model are also used by the other model. We make three contributions:

- RSVC, a new approach for providing human-interpretable insights into model differences (Fig. 1).

- A validation strategy to link representational differences to model decisions.

- Experiments to show that RSVC can measure similarity at both coarse- and fine-grained levels.

## 2 RELATED WORK

### 2.1 REPRESENTATIONAL SIMILARITY

Similarity methods attempt to quantify the similarity/dissimilarity between pairs of different models (Hotelling, 1936; Kornblith et al., 2019; Raghu et al., 2017; Li et al., 2015; Huh et al., 2024). Models can be compared based on their *functional* similarity (i.e., how their outputs relate) or their *representational* similarity (i.e., how the activations of intermediate layers relate) (Klabunde et al., 2023). While functional similarity can tell us about how model outputs vary, two models can achieve the same performance with significantly different representations. These differences matter, e.g., while two different forms of pretraining can achieve similar performance on certain datasets, they may transfer poorly to others (Xie et al., 2023).

Representational similarity metrics have successfully been used to analyze the differences between architectures (Nguyen et al., 2021; Raghu et al., 2021), explore the effects of different kinds of pretraining (Xie et al., 2023; Neyshabur et al., 2020; Park et al., 2024), develop novel strategies for efficient ensembling (Zhang et al., 2020), or perform ensembling that is robust to distribution shifts (Lee et al., 2023). Some methods have leveraged representational similarity to build tools for text-to-image generation (Rombach et al., 2020) or model-to-model translation (Dravid et al., 2023). In neuro- and cognitive science, representational similarity is used to measure how well models are able to approximate the neural recordings of the brain and/or the behavior of humans (Schrimpf et al., 2018; Muttenthaler et al., 2023; Fel et al., 2022b; Ahlert et al., 2024; Kriegeskorte et al., 2008). In the disentanglement literature, representational similarity has been used to evaluate how well the learned representation matches known ground truth latent factors. A popular approach is to use regularized linear predictors to map learned factors to ground truth latent factors (Eastwood & Williams, 2018; Eastwood et al., 2023; Locatello et al., 2019; Roth et al., 2023; Duan et al., 2020). Recently, Dravid et al. (2023) proposed a method that uses correlated activation patterns across networks to mine for "Rosetta Neurons". These neurons provide insights about features that re-occur consistently in many models. In contrast to our proposed method, Rosetta Neurons do not quantify the overall similarity between networks and do not identify neurons that explain model differences. Most closely related to our work is that of (Schrimpf et al., 2018) and (Li et al., 2015) which use a linear regression model as a similarity metric between the activations of two networks. We describe these methods in more detail in Sec. 3.2.

The primary limitation of most existing representational similarity methods is that they can quantify how similar the representations of two models are, but can not tell users *what* makes the models similar or dissimilar. We propose a new approach that aims to address this question. Our approach leverages concept-based explainability as an intermediate step to measure representational similarity.

### 2.2 EXPLAINABLE AI (XAI)

XAI methods aim to answer the questions (1) *what* features did a model use to arrive at a decision and (2) *where* is the relevant information in the input. Local explanation methods focus on pixel-based attribution, in which a heatmap indicates the region of the image that is most relevant to the model's decision (Selvaraju et al., 2020; Ribeiro et al., 2016; Lundberg & Lee, 2017). While local explanations are able to answer the "where" question, it can be challenging to interpret "what" is being highlighted as attributions can be noisy.

To better address the "what" question, global concept-based explanations can be used (Kim et al., 2018; Ghorbani et al., 2019; Zhang et al., 2021; Fel et al., 2023a; Kowal et al., 2024; Poeta et al., 2023; Bau et al., 2020). These approaches discover groups of images (or image regions) that share some visual feature that is relevant to the model's decision making. In addition, "glocal" methods have been developed to answer both questions simultaneously (Schrouff et al., 2021; Fel et al., 2023b; Achtibat et al., 2023; Kondapaneni et al., 2024).

Ilyas et al. (2022) take a different approach to explaining model decisions whereby linear surrogate models are trained to reproduce the output of a deep neural network. Then, each training image is given a score that reflects how much it contributed to the final weights of the learned model. In (Shah et al., 2023), this approach was extended to analyze the differences between two models by identifying the differences in training images that each model relies on to arrive at different decisions. While our work also aims to compare two models in an interpretable manner, our approach uses

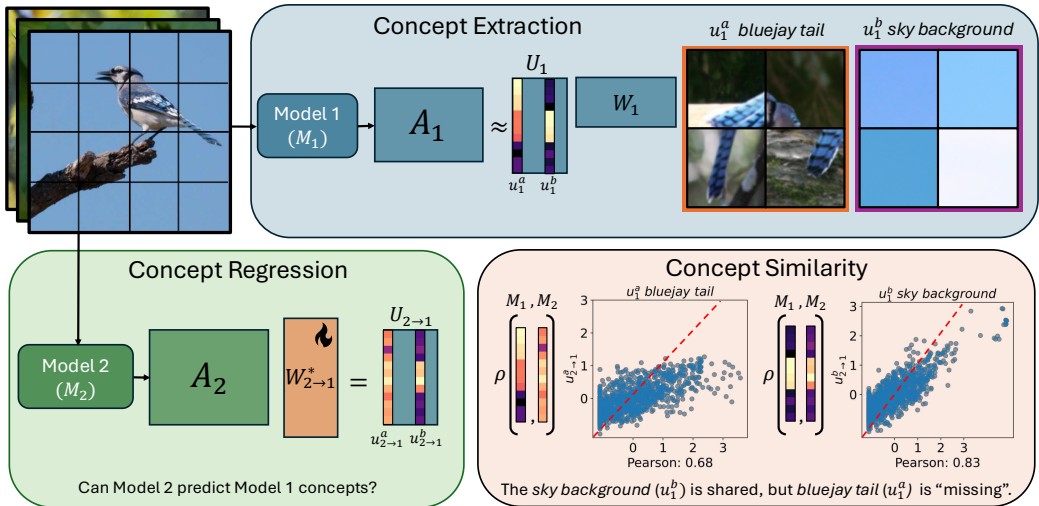

Figure 1: **Representational Similarity via interpretable Visual Concepts (RSVC).** (Concept Extraction): First, activations for a set of image patches, $\mathcal{I}^c$, are computed for each model ($M_1$ and $M_2$). Second, the activation matrix for $M_1$ is factorized into the *concept coefficient matrix* $\mathbf{U}_1$ and the *concept basis* $\mathbf{W}_1$, i.e., $\mathbf{A}_1 \approx \mathbf{U}_1\mathbf{W}_1$. Each entry in a column vector of the coefficient matrix $\mathbf{U}_1$ represents the strength of a concept in an image. Concepts are visualized by the image patches that correspond to the top $n$ coefficients. Here, we highlight only two concepts, $u_1^a$ and $u_1^b$. The top four images for these concepts indicate that $u_1^a$ represents *bluejay tail* and $u_1^b$ represents *sky background*. (Concept Regression): To measure concept similarity, we learn a weight matrix $\mathbf{W}_{2\rightarrow1}^*$ to map $\mathbf{A}_2$ to the concept coefficient matrix $\mathbf{U}_2$. We denote the predicted coefficient matrix as $\hat{\mathbf{U}}_{2\rightarrow1}$. (Concept Similarity): Finally, we compute the correlation between columns of $\hat{\mathbf{U}}_{2\rightarrow1}$ and $\mathbf{U}_1$. If $\mathbf{A}_2$ contains a concept in $\mathbf{U}_1$, then the predicted coefficient vector should be highly correlated to the real coefficient vector. In this example, we see that the *bluejay tail* concept is poorly represented in $M_2$, but both models share the *sky background* concept.

the activations of the original model itself. This gives RSVC the advantage of being able to link representational changes to functional model behavior.

## 3 METHOD

We propose a new approach to compare the representations of two models using concepts. Our approach is closely related to methods for computing representational similarity, such as Brain-Score (Schrimpf et al., 2018) and the metric in (Li et al., 2015), and builds upon prior work in concept extraction (Fel et al., 2023b; Ghorbani et al., 2019; Fel et al., 2023a). We bridge approaches from these two fields, resulting in an interpretable method to measure representational similarity for deep neural networks (Fig. 1).

### 3.1 CONCEPT EXTRACTION

Fel et al. (2023a) showed that many concept based explainability approaches can be generalized as dictionary learning methods. In these approaches, a set of $n$ input images $\mathcal{I}$ is used to compute activations from a specific layer $l$ of a neural network resulting in an activation matrix $\mathbf{A} \in \mathbb{R}^{n \times d}$, where each row is an activation vector of dimensionality $d$. Then, a dictionary learning algorithm can be used to approximate the activations as $\mathbf{A} \approx \mathbf{U}\mathbf{W}$. The row vectors of the vector basis $\mathbf{W} \in \mathbb{R}^{k \times d}$ can be interpreted as a set of $k$ concepts. Similarly, the rows of the coefficient matrix $\mathbf{U} \in \mathbb{R}^{n \times k}$ represent the importance of a particular concept vector in $\mathbf{W}$ for a given image. By visualizing the images that have the largest concept coefficients for a given concept, we are able to understand *what* visual concepts the network has identified in the data. The effectiveness of such concept based XAI methods has been demonstrated via user studies in previous work (Fel et al., 2023b; Ghorbani et al., 2019; Kim et al., 2018).

Our goal is to develop a method to compare the representational similarity between two models, not just in terms of a single numerical score, but via interpretable concepts. For each model, we use the concept extraction approach proposed in CRAFT (Fel et al., 2023b). We denote the first model as $M_1$, and the second as $M_2$. Importantly, we make no assumption that these two models are from the same model family, e.g., one could be a CNN and the other a Vision Transformer. We outline the concept extraction process for $M_1$, but the same process is applied to $M_2$.

For a specific object class $c$, we select the set of images $\mathcal{I}_1^c$ that $M_1$ predicted to contain class $c$. By grouping images according to model predictions, as opposed to only the ground truth labels, we are able to identify concepts used in all images that the model believes are part of class $c$. This allows the method to provide better insight into both correct and incorrect predictions. Images are usually composed of several visual concepts, so we extract evenly spaced patches from the image to form a set of "concept proposal" images. These patches are more likely to contain a single visual concept and are easier to interpret. We re-use $\mathcal{I}_1^c$ to refer to the set of concept proposal images.

Each concept proposal is resized to the model's input resolution and passed through the network. Note that all networks are trained with cropping augmentation, such that patches are in the domain of training images. We denote the activations from a specific layer $l$ and class $c$ as $\mathbf{A}_1 \in \mathbb{R}^{|\mathcal{I}_1^c| \times d}$, where $d$ is the dimension of the activations of the layer. We then use a dictionary learning algorithm with $k$ components to decompose $\mathbf{A}_1$ into a matrix $\mathbf{U}_1 \in R^{|\mathcal{I}_1^c| \times k}$ and $\mathbf{W}_1 \in \mathbb{R}^{k \times d}$, such that $\mathbf{A}_1 \approx \mathbf{U}_1 \mathbf{W}_1$. We refer to $\mathbf{U}_1$ as the *concept coefficient matrix* and $\mathbf{W}_1$ as the *concept basis*. We repeat this process for $M_2$, resulting in $\mathbf{U}_2$ and $\mathbf{W}_2$. Intuitively, each row of a concept coefficient matrix $\mathbf{U}$ encodes the contribution of each concept vector in $\mathbf{W}$ to the activation vector of a particular image in $\mathcal{I}$.

To measure concept similarity between two models, we need to understand how each concept reacts to the *same* set of images. For example, if both networks have discovered a concept that reacts to the color red, they would both have an activation pattern that spikes when red objects are presented. Following this logic, we propose that if two concept vectors are encoding the same information, their concept coefficients over the same set of images should be correlated. To obtain a shared set of images, we take the union over the image sets $\mathcal{I}_1^c \cup \mathcal{I}_2^c$ to form $\mathcal{I}^c$. The proposals are passed through the model $M_1$ to produce $\mathbf{A}_1$ for the shared concept set. Given the concept basis $\mathbf{W}_1$ (which is specific to $M_1$), we re-compute $\mathbf{U}_1$ over the shared set of images. We repeat this process for $M_2$ to compute $\mathbf{U}_2$. In practice, we use a non-negative least squares solver, since the original coefficient matrices are non-negative (Appendix B).

## 3.2 Concept Similarity

Here we address the following question: in a specific pair of layers, does $M_1$ encode the same concepts as $M_2$? In the following sections, we make use of $\mathbf{A}_1$, $\mathbf{A}_2$, $\mathbf{U}_1$ and $\mathbf{U}_2$ to compute the similarity between concepts encoded in $M_1$ and $M_2$. We consider two different approaches that trade-off computational cost and error. In Appendix A.1 we describe a correlation based metric that has a low computational cost that we use to coarsely compare many layers of two neural networks.

In order to more accurately measure similarity in a single layer, we propose a regression based metric similar to the strategy in (Li et al., 2015) and the BrainScore (Schrimpf et al., 2018), which was originally introduced to compare artificial neural networks to biological neural networks. In (Li et al., 2015), the outputs of a convolutional layer in one model are mapped to the outputs of a convolutional layer in another network using a sparse weight matrix. The prediction error between the predicted outputs and true outputs are used as a metric for the similarity between the two layers. In BrainScore, for a set of $n$ stimuli (e.g., images), the activations from a layer of the DNN are stored in a matrix $\mathbf{A} \in \mathbb{R}^{n \times d}$, where $d$ is the dimensionality of the layer activations. For the same set of stimuli, neural recordings are measured from an animal and processed forming a vector $\mathbf{y} \in \mathbb{R}^n$ for each target "neuroid". A linear mapping is introduced to predict the neural responses from the DNN activations, $\mathbf{y} = \mathbf{A}\mathbf{w}^* + b$, where $\mathbf{w}^*$ are the weights of the regressor and $b$ is the bias. The regression model is trained on a set of training images and evaluated on held out test images where the predicted outputs of the regressor are compared to the true neural responses using Pearson correlation, giving a score between -1 and 1.

To compare two neural networks using these methods, each column of $\mathbf{A}_2$ would serve as prediction targets for regression with $\mathbf{A}_1$ resulting in a similarity score for each column of $\mathbf{A}_2$. However,

visualizing and interpreting each neuron results in an explanation that is too complex for users. Instead, a more interpretable result can be achieved by setting the coefficient matrix $\mathbf{U}_2$ as regression targets for $\mathbf{A}_1$ (Fig. 1). Essentially, RSVC encodes similarity by measuring how well $M_1$ can predict the concept coefficients of $M_2$ and vice-versa,

$$\mathbf{A}_1 \mathbf{W}^*_{1 \to 2} = \mathbf{U}_{1 \to 2} \quad \text{and} \quad \mathbf{A}_2 \mathbf{W}^*_{2 \to 1} = \mathbf{U}_{2 \to 1}, \tag{1}$$

where we learn $\mathbf{W}^*$ such that the following regularized ($l_1$) mean-squared error is minimized.

$$\min_{\mathbf{W}^*} \frac{1}{n} \sum_{i=1}^{n} (\mathbf{A}\mathbf{W}^* - \mathbf{U})^2 + \lambda \|\mathbf{W}^*\|_1. \tag{2}$$

$l_1$ regularization guides the regression model to seek a sparse set of neurons in $M_1$ that can be used to predict $U_2$ and reduces over-fitting to the regression training data. We also compute baselines for each model from their own activation matrices, learning $\mathbf{W}^*_{1 \to 1}$ and $\mathbf{W}^*_{2 \to 2}$. Finally, we compute the Pearson and Spearman correlation between columns of the predicted coefficient matrix and columns of the true coefficient matrix to get a similarity score for each concept between -1 and 1. We refer to the score as cross-model concept similarity (CMCS) when computed across two models, and as same-model concept similarity (SMCS) when computed within the same model.

Finally, we investigate whether similar or dissimilar concepts are more important to model decisions by applying concept integrated gradients (Fel et al., 2023a). Concept integrated gradients measure the contribution of each concept to a model's decision (Appendix B.1.1).

## 3.3 REPLACEMENT TEST

To link differences in predicted concept coefficients and real concept coefficients to model behavior, we conduct a "replacement test". For each model comparison, we conduct a replacement operation over each column of the coefficient matrix and keep track of the resultant reconstructed activation matrix, model logits, and model predictions (as seen in the pseudocode on the left). We perform the replacement operation using the same model predicted coefficients $\mathbf{U}_{1 \to 1}$ (baseline) and also the cross-model predicted coefficients $\mathbf{U}_{2 \to 1}$. We measure the impact of replacement at three levels during a classification task. We denote $h$ as the classification head that produces logits ($\mathbf{z}$) for each input image. We compute the mean $l_2$-distance between each row of $\bar{\mathbf{A}}^i_{2 \to 1}$ and $\bar{\mathbf{A}}^i_{1 \to 1}$, the mean KL-divergence over the logits $\bar{\mathbf{z}}^i_{2 \to 1}$ and $\bar{\mathbf{z}}^i_{1 \to 1}$, and the match accuracy between $\bar{\mathbf{y}}^i_{2 \to 1}$ and $\bar{\mathbf{y}}^i_{1 \to 1}$.

```
1: for each i = 1 to K do
2:    Ū^i_{2→1} = Copy U_1
3:    Ū^i_{2→1}[:, i] = U_{2→1}[:, i]
4:    Ā^i_{2→1} = Ū^i_{2→1} W_1
5:    z̄^i_{2→1} = h(Ā^i_{2→1})
6:    ȳ^i_{2→1} = arg max(z̄^i_{2→1})
7: end for
```

## 3.4 INTERPRETING LOW SIMILARITY CONCEPTS

How should we visually compare predicted and real concepts? Concept based XAI methods like CRAFT (Fel et al., 2023b) and CRP (Achtibat et al., 2023) visualize the $n$ images with the largest concept coefficient as representatives of the visual feature encoded by the concept. The same approach can be used to visualize similar concepts, since, by definition, similar concepts have highly correlated activation patterns over the shared set of concept proposal images. However, this approach is misleading when low similarity concepts are discovered. When a concept is dissimilar it may share the same top $n$ images, but have entirely uncorrelated coefficients over the remaining images. This possibility is further amplified due to the mean squared error (MSE) loss used to estimate the regression matrix $\mathbf{W}^*_{1 \to 2}$. The MSE loss penalizes prediction error on the largest coefficients disproportionately, leading to a higher chance the two models share the same top $n$ images.

To address this, we develop a new approach to specifically visualize how two models are dissimilar with respect to a concept. We start by visualizing the target concept by using an image collage corresponding to the top $n$ concept coefficients. The target concept is compared to the predicted concept by visualizing the top $n$ over-predicted coefficients and top $n$ under-predicted coefficients. This allows users to reason about visual features that one model entangles or disentangles with the target concept, improving their overall understanding of the compared models. To ensure sample diversity, we enforce that we visualize one patch per image. We also exclude the top-10 real concept images when selecting the over and under predicted points. In Figs. 2 and 5, we demonstrate our proposed approach for interpreting the dissimilarity of the concepts. We include more visualizations in Appendix A.2.

### 3.5 IMPLEMENTATION DETAILS

We choose a ResNet-18 (RN18), ResNet-50 (RN50) (He et al., 2016), ViT-S, ViT-L (Dosovitskiy, 2021), DINO ViT-B (DINO) (Caron et al., 2021), and MAE ViT-B (MAE) (He et al., 2022) from the timm library (Wightman, 2019) for our experiments. All models were trained on ImageNet (Deng et al., 2009). For our exploration with DINO and MAE, we finetune the models on NABirds (Van Horn et al., 2015). We compare four pairs of models: RN18 vs. RN50, ViT-S vs. ViT-L, RN50 vs. ViT-S, and DINO vs. MAE. Model performance on their respective datasets are reported in Tab. A1. The first two pairs have a clear difference in performance, implying that there are significant representational differences between the models. The second two pairs are roughly equal in overall performance, allowing us to explore how representational differences may result in different behavior even when overall performance is the same. For the $l_1$ penalty, we sweep $\lambda$ on a subset of data and find $\lambda = 0.1$ to be a reasonable choice (Fig. A13). Additionally, we set the number of concepts $k = 10$ to balance reconstruction error and computational cost (Fig. A14). We provide further details on the concept extraction, concept comparison and computational cost in Appendix B.

## 4 RESULTS

### 4.1 DISCOVERING A "TOY" CONCEPT

While understanding models on real data is closely related to real-world use cases, it results in complex concepts that can be more challenging to interpret. To better understand the properties of RSVC, we design an experiment in which we train a model on images modified with a simple toy visual concept. In this experiment, we are able to control what the model learns and explore how RSVC works in a more controlled setting.

We train two ResNet-18 models from scratch on all classes from a modified NABirds dataset (Van Horn et al., 2015). The first model, $M_{ps}$, is trained to make use of the toy concept in its decisions and the second, $M_{nc}$ is trained to become invariant to the toy concept. The toy concept is a $20px \times 20px$ pink square that is stochastically placed on the images at a random location. For $M_{ps}$, the concept appears on images from the Common Eider class with a 70% probability, giving the concept predictive power. For $M_{nc}$, the concept appears on images from any class with a 50% probability, giving the concept no predictive value. Thus, $M_{ps}$ should learn to attend to the visual concept while $M_{nc}$ should learn to ignore the concept, since it is simply noise. Finally, both models are tested on a dataset in which the concept appears on images from the Common Eider class with a 100% probability. Provided that $M_{nc}$ successfully learns to ignore the concept, RSVC should have a low similarity score to any concept in $M_{ps}$ that primarily fixates on the pink square. Recall that concepts are visualized using image patches. This allows us to break the 100% correlation between the pink square and the image during testing, such that only some patches contain the added concept. Both models achieve $\sim 34\%$ classification accuracy on the test set of NABirds.

In Fig. 2, we visualize the similarity from $M_{nc}$ to Concept 1 in $M_{ps}$. We find that $M_{nc}$ has a near zero similarity to Concept 1 in $M_{ps}$, which we visually identify to be a concept that fixates on the pink square. Importantly, the modified training paradigm does not affect the similarity scores between other concepts in the two models. In Fig. A5, we show that $M_{nc}$ has high similarity to two other concepts in $M_{ps}$. Thus, we show that RSVC clearly identifies the primary conceptual difference between the two models in this controlled experiment.

### 4.2 CONCEPT SIMILARITY VS. CONCEPT IMPORTANCE

To analyze real data, we start by exploring the relationship between concept similarity and importance in the penultimate layers of different models trained on ImageNet (Deng et al., 2009). We compute the cross-model concept similarity (CMCS) and the concept importance (CI) for every extracted concept. In Fig. 3, we plot the concept similarity from $M_1$ to $M_2$ against the concept importance for $M_2$ across four pairs of models. In this figure, a point with low similarity and high importance would indicate that in layer $l$, $M_1$ can *not* predict the coefficients of a concept that $M_2$ finds important in decision making. We use color to indicate the density of points in a region (warmer colors indicate more density). In Appendix A.7, we compare same-model concept simi-

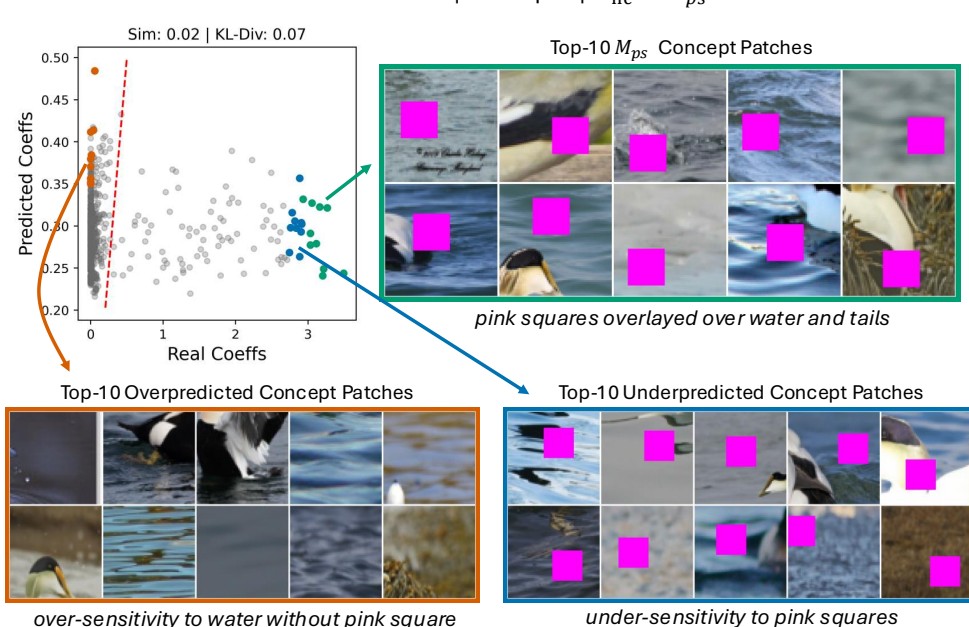

Figure 2: **Adding and Discovering a Toy Concept.** Here we train two ResNet-18 models, $M_{ps}$ and $M_{nc}$. $M_{ps}$ is trained to associate a pink square (i.e., Concept 1) with the Common Eider class, while $M_{nc}$ is trained to be invariant to the pink square concept. We find that the similarity score from $M_{nc} \rightarrow M_{ps}$ for Concept 1 is $\sim 0.0$, indicating that $M_{nc}$ is unable to predict Concept 1 from $M_{ps}$. To understand various aspects of the differences between the two models, RSVC inspects three distinct regions of the predicted vs. real coefficient scatter plot (Sec. 3.4). (Green): RSVC visualizes images corresponding to the top-10 $M_{ps}$ target concept coefficients. This allows the user to understand what the target concept is encoding. This concept clearly reacts strongly to the pink square visual feature. (Blue): RSVC visualizes the image patches with the largest $M_{nc}$ under-predicted coefficients. $M_{nc}$ under-reacts to the pink square when compared to $M_{ps}$. (Orange): RSVC visualizes the image patches corresponding to the top-10 $M_{nc}$ over-predicted coefficients. The over-predicted patches show that $M_{nc}$ cannot distinguish between background and the pink square.

larity (SMCS) to CI. As expected, we find that SMCS values are significantly higher than CMCS values, since the model is predicting its own concepts.

In cross-model comparisons, we observe that models tend to have medium/high similarity for most concepts, since dense regions in the plot tend to be above 0.6 similarity. We also notice that the similarity scores from ViTs and ResNets have a different overall structure, with ResNets having a longer tail of low similarity and low importance concepts. Finally, except for DINO vs. MAE we find that there are several low/medium similarity concepts that also have a medium/high importance. In Appendix A.5, we systematically vary the training protocol (seed and data) for a ResNet-18 model and measure the impact of the changes on model similarity. We find that changes in model training lead to intuitive changes in similarity and use RSVC to reveal some concepts that suggest how the two models differ (Fig. A7). In summary, we observe two key results: (1) model differences are largely driven by medium similarity, medium importance concepts, jointly contributing to significant changes in model behavior and (2) some models do learn "unique" low similarity, high importance concepts. In the following sections we explore both of these results further.

## 4.3 REPLACEMENT TEST

In order to better understand how variations in similarity impact model behavior we conduct a replacement test (described in Sec. 3.3). This test allows us to measure how changes in concept similarity impacts the $l_2$-distance of the activations, KL-divergence of the logits, and match accuracy of the predictions. We investigate this question because it is possible that changes in Pearson correlation are due to changes in predicted coefficients on unimportant images for a particular concept,

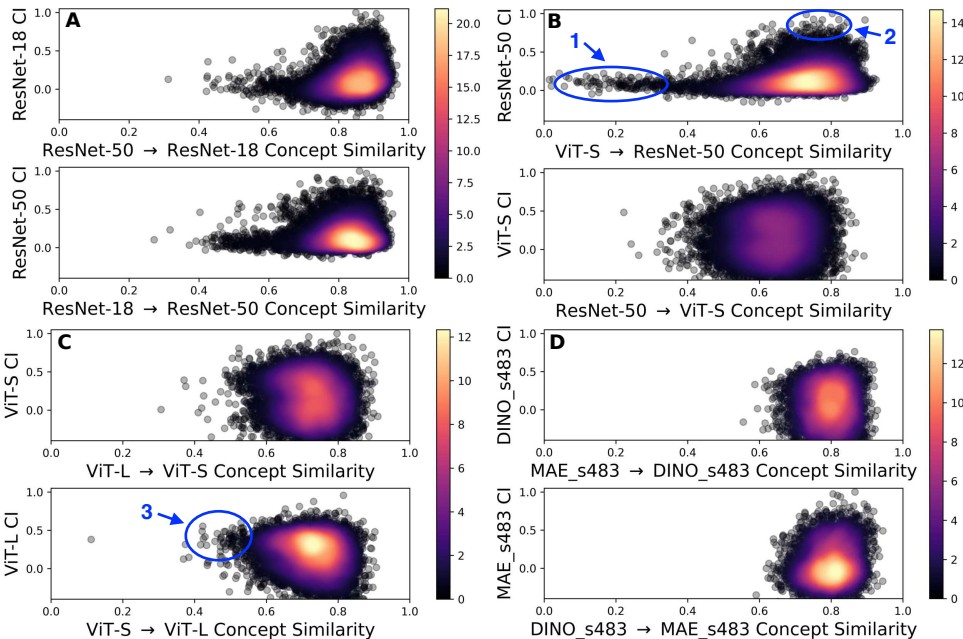

Figure 3: **Concept Similarity vs. Concept Importance.** We compare four pairs of models using CMCS: (A) RN18 vs. RN50, (B) RN50 vs. ViT-S, (C) ViT-S vs. ViT-L, and (D) DINO vs. MAE. The y-axis represents the concept importance (CI) measured using concept integrated gradients. Warmer colors represent the density of points in a region. We highlight several regions in the plots: (1) low similarity and low importance concepts that are unique to a model but contribute little to its decisions, (2) high importance and high similarity concepts that are shared across both models and also contribute greatly to decision making, (3) low similarity, high importance concepts that only one model has discovered, but are very important to that model's decisions.

leading to no change in model behavior. In Fig. 4, we visualize the change in Pearson correlation ($\Delta$Pearson) against the change in the three aforementioned metrics. We use color to indicate concept importance (warmer colors are more important). As expected, we observe a trend showing that $l_2$-distance increases as similarity decreases. For the KL-divergence, we observe two trends: (1) when the importance is sufficiently high, the KL-divergence increases as similarity decreases and (2) when the importance is low, there is no effect on the model's logits. Finally, we observe a trend that shows that model predictions change as a function of both similarity and importance. We find that these trends roughly hold for all models, although the structure of the plots changes for ViTs. See Appendix A.8 for more results.

## 4.4 LOW SIMILARITY CONCEPTS

In Fig. 3 we observe that model comparisons identify low similarity, high importance concepts. These concepts are particularly interesting because they identify visual features that one model has constructed that the other has not. In Fig. 5 we apply our proposed approach for understanding the dissimilarity between predicted and real concepts (see Sec. 3.4). We analyze a RN50 concept used on the barbell class, which primarily reacts to images containing hands/arms lifting a barbell. When a regression model is trained on the ViT-S activations to predict the coefficients of the RN50 concept, the model becomes over-sensitive to any image with weight plates or gym equipment and remains under-sensitive to images of people around gym equipment. This result suggests that the ViT-S does not have a feature encoding "hands lifting weights". In Fig. A3, we show some more examples of important low similarity concepts that are visually interpretable.

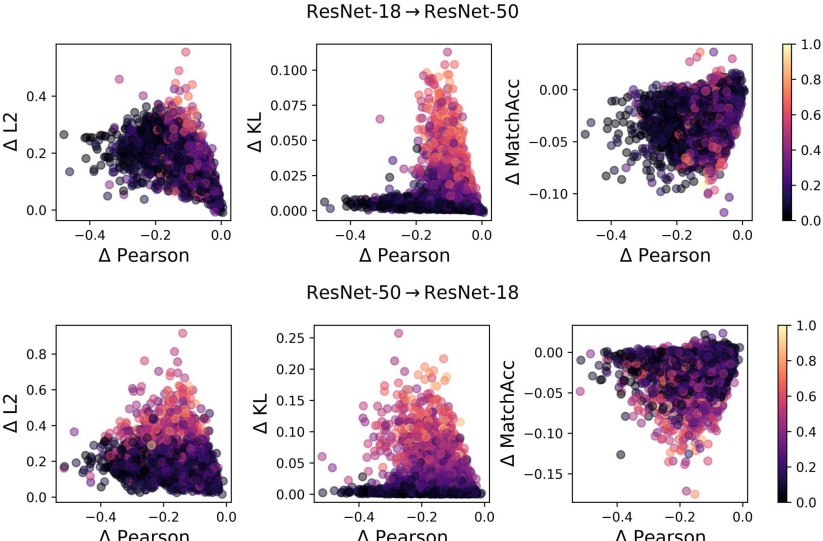

Figure 4: **Replacement Test.** We determine whether poorly predicted coefficients for concepts actually impact model behavior (Sec. 4.3). We use color to represent the concept importance (warmer is higher importance). When ignoring low importance concepts, we observe expected trends, i.e., decreases in similarity ($\Delta$Pearson) result in increases in the $l_2$-distance, increases in KL-divergence on the classifier logits, and decreases in model accuracy. The effect also seems to be scaled by importance, for example, changes to low importance concepts (black) has no impact on $\Delta$KL.

## 4.5 LLVM GENERATED CONCEPT DIFFERENCE ANALYSIS

Fig. 5 contains a manual analysis of the difference between a visual concept for the barbell class from two different models. Here we use an LLVM (ChatGPT-4o Achiam et al. (2023)) to analyze image collages and describe them. This approach has a similar goal to Gandelsman et al. (2024), in which the authors use an LLVM to interpret specific MLP heads in a CLIP ViT model. The input and output structure for this experiment is provided in Fig. A4. We do not emphasize differences in the system prompt, instead we ask the model to describe both similarities and differences. We provide the top-k image collage (IC1) and the over-predicted image collage (IC2) as input to the LLVM. The LLVM outputs for the barbell concept are presented below. We observe that the LLVM description is similar to the manual description in the caption:

---

**LLVM Analysis of a Visual Concept from the Barbell class**

**IC1**: A collage of people engaging in strength training exercises, with a focus on lifting dumbbells, barbells, and performing weightlifting movements.
**IC2**: A collage showcasing gym equipment and weights, with some emphasis on exercises involving barbells and dumbbells but with fewer people depicted.
**Similarity**: Both collages center on weightlifting and gym-related activities, featuring equipment and exercises.
**Difference**: IC1 highlights people actively performing weightlifting exercises, while IC2 primarily focuses on the gym equipment and setup with minimal human involvement.
**Semantically different**: [Yes] IC1 emphasizes the activity of weightlifting, whereas IC2 focuses more on the tools and environment of the gym.

---

## 5 LIMITATIONS

We limit our analysis to computer vision models. In practice, there is nothing restricting the application of the RSVC approach to large language models, but we leave this for future work. Our work builds on concept-based XAI methods, and thus inherits the trade-off between fidelity and interpretability. In particular, we note that the reconstruction error varies for each model, which may explain some properties of concept similarity in our experiments. However, we are agnostic to the precise concept extraction method used and thus our approach will benefit from further ad-

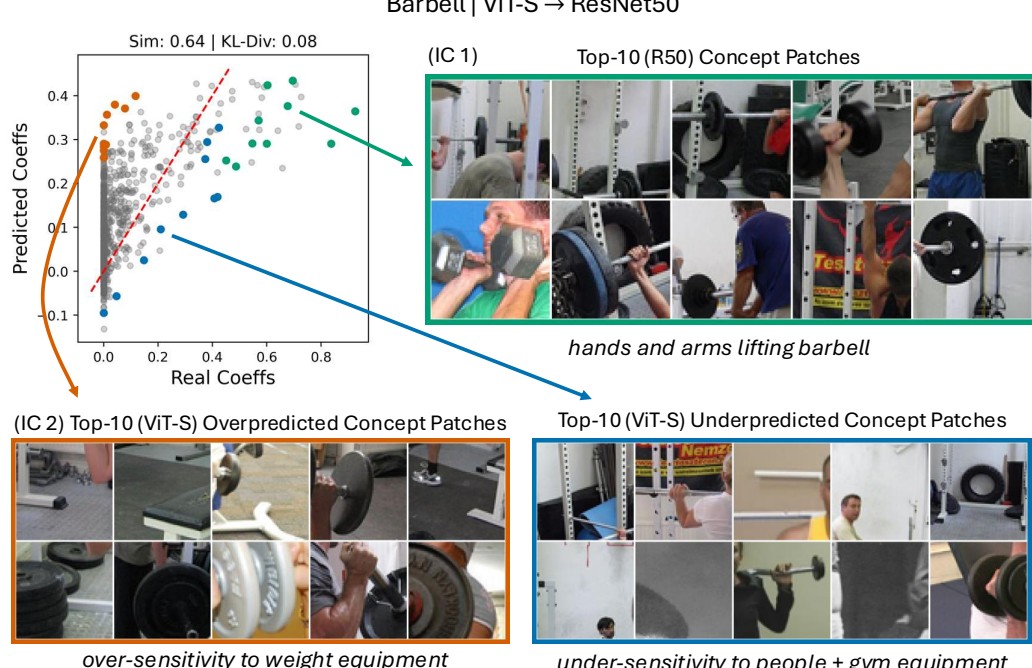

Figure 5: **Interpreting Low Similarity Concepts.** In this example, we find a RN50 concept for the barbell class that the ViT-S is not able to predict. (Green): The RN50 concept reacts to images of hands lifting barbells. Additionally, many images contain vertical supports for a squat rack. We train a regression model on the ViT-S activations to predict the RN50 concept coefficients. (Blue): The ViT-S regression model under-reacts to images containing hands, people, and squat racks. (Orange): It over-reacts to images that have a greater focus on weight plates. These results suggest that the the specific concept of hands lifting barbells is not represented in the ViT-S. In Sec. 4.5 we use an LLVM to analyze the image collages (IC1 and IC2) and find that it detects similar differences in the visualizations.

vances in these methods. For example, incorporating a recursive strategy like the one presented in CRAFT (Fel et al., 2023b), may significantly improve the number of interpretable comparisons discovered. Outside of overly artificial settings whereby two models are trained on completely different datasets, we note that it can be challenging to compare the representations of two models trained on the same or similar datasets (e.g., ImageNet) as done in this work. However, we believe that this more challenging setting is of most interest and relevance.

## 6    CONCLUSION

We introduced a new method for exploring representational similarity via interpretable visual concepts (RSVC). In contrast to existing representational similarity methods that simply provide a single numerical score to denote similarity, our approach fuses ideas from concept-based explainability and shows us *what* visual concepts make two models similar or dissimilar. In particular, we demonstrate that comparing models can be an effective path towards understanding what concepts a model is missing. In future, this may be helpful in identifying sources of model failures. We presented experiments on a range of different vision models and demonstrated that our approach is general and can be applied across a variety of different backbone models, irrespective of the pretraining objective used by the model. Finally, we suggest that explaining the functional differences in two models' behavior could serve as a valuable testbed for future XAI research. We hope that our work opens the door to further investigation into how concepts are represented inside of deep networks.

ACKNOWLEDGMENTS

NK and PP were supported by the Simons Foundation and the Resnick Sustainability Institute. OMA was supported by a Royal Society Research Grant. The authors thank Rogério Guimares, Markus Marks, Atharva Sehgal, and the anonymous reviewers for their valuable feedback.

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

# Appendix

## A  ADDITIONAL EXPERIMENTS

### A.1  LAYERWISE CONCEPT SIMILARITY

#### A.1.1  CORRELATION

We propose a correlation based comparison method that has a lower computational cost than the regression method presented in the main text (Sec. 3.2). This approach allows us to efficiently compare many concepts and layers for two models. Recall that each column of $\mathbf{U}$ contains the concept coefficients of a specific concept for each image. If the concepts encode the same information, they should have highly correlated activation patterns since they would react similarly to the same proposal images. Thus, we compute the correlation for each vector $\mathbf{u}_1^i \in \mathrm{Columns}(\mathbf{U}_1)$ and $\mathbf{u}_2^j \in \mathrm{Columns}(\mathbf{U}_2)$. We measure both Pearson and Spearman correlation to form the correlation matrices $\mathbf{R}^\rho \in \mathbb{R}^{k \times k}$ and $\mathbf{R}^S \in \mathbb{R}^{k \times k}$. Since concepts extracted from each network do not have a direct correspondence, MCS measures the concept similarity between a concept from $M_1$ and all of the concepts from $M_2$ and keeps the maximum. We compute the maximum concept similarity ($\mathrm{MCS}^c$) over each dimension of the correlation matrix for each concept and class

$$\mathrm{MCS}_1^c = \max_i \mathbf{R}_{ij} \quad \text{and} \quad \mathrm{MCS}_2^c = \max_j \mathbf{R}_{ij}. \tag{3}$$

MCS is fast to compute and can be done with relatively few image samples (Appendix B.2). Thus, we use it to compute layerwise concept similarity which gives us coarse-grained insights into the similarity between each layer of two different models. For layerwise concept similarity, we compute the correlation matrix $\mathbf{R}^c$ between $M_1$ and $M_2$ at each pair of layers for every class $c$. Then, for each correlation matrix we compute the *mean* maximum concept similarity (MMCS) over each dimension and then take the average of the two matrices

$$\mathrm{MMCS}_m = \frac{1}{k \cdot c} \sum_{c=1}^{c} \sum_{i=1}^{k} \mathrm{MCS}_m^{c,i},$$

$$\mathrm{MMCS} = (\mathrm{MMCS}_1 + \mathrm{MMCS}_2)/2. \tag{4}$$

However, correlation based similarity can be affected by confounds from concept extraction. For example, extracted concepts can entangle or disentangle visual features that are encoded by the respective models. Suppose $\mathbf{U}_1$ encodes features for both ears and snouts of a dog together in a single concept, but these two features are disentangled into two concepts in $\mathbf{U}_2$, the maximum match between these concepts would be lower even though both networks are encoding the same information. Additionally, extracted concepts do not contain all the information in the network since there is some reconstruction error when learning the decomposition. Thus, correlation matrices can tell us when two concepts are highly similar, but do not tell us if a concept is missing in a layer. However, they can serve as a noisy lower bound for measuring concept similarity (Appendix A.1.3).

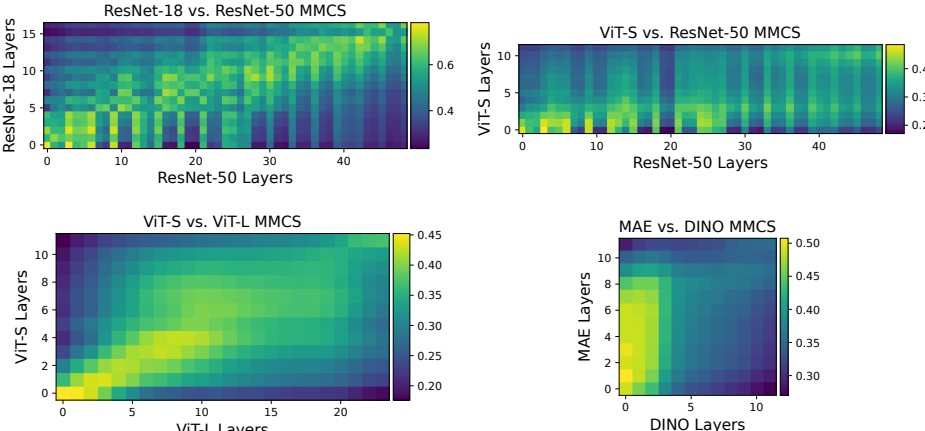

Figure A1: **Layerwise Mean-Max Concept Similarity.** We compare four pairs of models across many selected layers using Pearson correlation. Each entry in the matrix is the mean maximum concept similarity (MMCS) between $M_1$ and $M_2$ at a particular pair of layers. Brighter colors represent higher MMCS values. We see that, in general, concept similarity is highest in earlier layers and decays as networks get deeper. We also notice that there is a slight increase in similarity towards the final layers (Appendix A.1).

### A.1.2 COMPARING MODEL LAYERS

In Fig. A1, we explore how concept similarity arises across many layers of each network. We compute the MMCS at each pair of layers and visualize the resulting matrix. For all models, we find that concept similarity is highest at the early layers of each model and decays gradually as network depth increases. In all model comparisons, we see a slight increase in similarity towards the final layers relative to the preceding layers, suggesting that the way networks organize information converges as the network get closer to producing a final decision. Interestingly, (Fel et al., 2022b) also found the last layer to have better properties for concept extraction.

We also notice several properties unique to each comparison. When comparing different sized models with the same architecture, the similarity is related to the relative depth of the layer. For ResNets, we notice that matrices show a pattern of increased similarity after residual blocks and lower similarity for layers within blocks. For the ViT-S and ViT-L we find that there is a broad band of concept similarity in the middle of each network in which the ViT-S layers 4 through 9 have higher similarity to ViT-L layers 5 through 20. In addition, there is an increase in concept similarity between the last layer of the ViT-S and the last three layers of the ViT-L. When comparing the RN50 to ViT-S we find that concept similarity of layers 0 through 25 are most similar to layers 0 through 3. This finding matches observations found in previous work, in which relatively earlier layers in the ViT match relatively later layers in the ResNet (Raghu et al., 2021). Finally, when comparing the MAE to DINO, we see high concept similarity between the first 3 layers of DINO and the first 8 layers of the MAE. However, this similarity decays significantly as the layer index of DINO increases. This divergence in concept similarity may be due to differences between supervised and self-supervised training, but further research is needed.

### A.1.3 COMPARING MCS (PEARSON) TO LASSO REGRESSION (PEARSON)

In Appendix A.1.1, we claimed that MCS (Pearson) could serve as a noisy lower bound to better measurements of similarity like CMCS (Pearson). This is because MCS is computed over concepts extracted by the decomposition method, which introduces its own entanglements, disentanglements, and reconstruction error. In Fig. A2, we compare the penultimate layers of the four pairs of models and compute both MCS (Pearson) and the CMCS (Pearson). We find that MCS is correlated to CMCS, but, as expected, under-predicts the concept similarity. For this experiment only, due to the fact that we computed Pearson correlation over images in the training set of the deep neural network, we compare to the mean similarity score that is computed from the held-out folds of the five lasso regression models. The held-out folds are not part of the training set for the *regression models*, but

were part of the training set of the deep neural network. However, this comparison is more fair, since we compare the two methods on the same set of images.

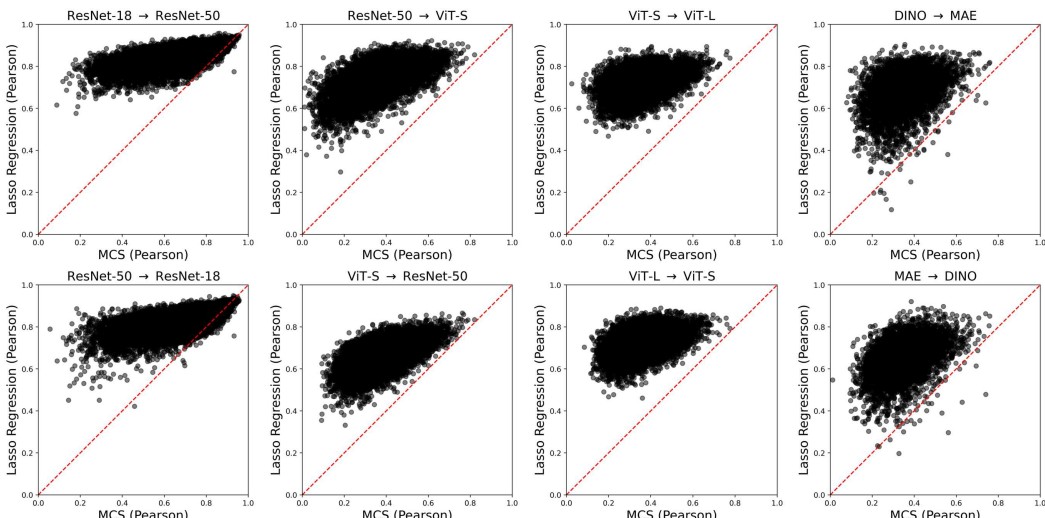

Figure A2: **MCS (Pearson) vs. Lasso Regression.** We see that the most points lie above the red-line. This means that lasso regression (followed by Pearson correlation on the predicted and real coefficients) usually predicts a higher similarity value than the MCS values directly on the columns of the coefficient matrix. Thus, we experimentally validate that the Pearson correlation acts as a noisy lower bound on concept similarity.

## A.2 ANALYZING CONCEPTS

In Fig. A3 we display some more examples of dissimilar concepts found through RSVC. We also provide interpretations of the dissimilarities between the concepts. Note that these dissimilarities do not necessarily indicate worse performance on images from this class because there are many possible correct strategies when trying to make a decision about an image.

### Rugby Ball | RN18 → RN50

The RN50 has learned a visual concept that entangles the arms of a rugby player and the rugby ball. We see in the under-predicted samples that the regression model under-predicts samples with the players' limbs and rugby balls together. When visualizing the over-predicted samples we can see that the regression model increases sensitivity to both close-ups of rugby balls and to the legs of the players. These results suggest that the RN18 encodes for the legs of the rugby players independently of the rugby ball and the regression model is using these independent features to try and reproduce the RN50's concept. It also suggests that the RN18 is not encoding the arms as an independent feature.

### Grey Whale | ViT-S → ViT-L

It appears that the ViT-L has learned a visual concept for whales surfacing parallel to the surface of the water. We can see that the regression model under-predicts images of the whales back and also images of its eye in a horizontal orientation. The regression model seems to have increased sensitivity to breaching whales, either raising their tails or their heads. This suggests that the ViT-S has entangled calm whales floating at the surface with active whales breaching the water.

### Strawberry | RN50 → ViT-S

This concept is one of the lowest similarity concepts that causes a meaningful change in model behavior (as seen by the KL-Divergence). We can see that the ViT-S has learned a concept for mixed fruits that include strawberries. The under-predicted and over-predicted samples show that the RN50 has no ability to reproduce this pattern, suggesting that it ignores mixed fruits entirely.

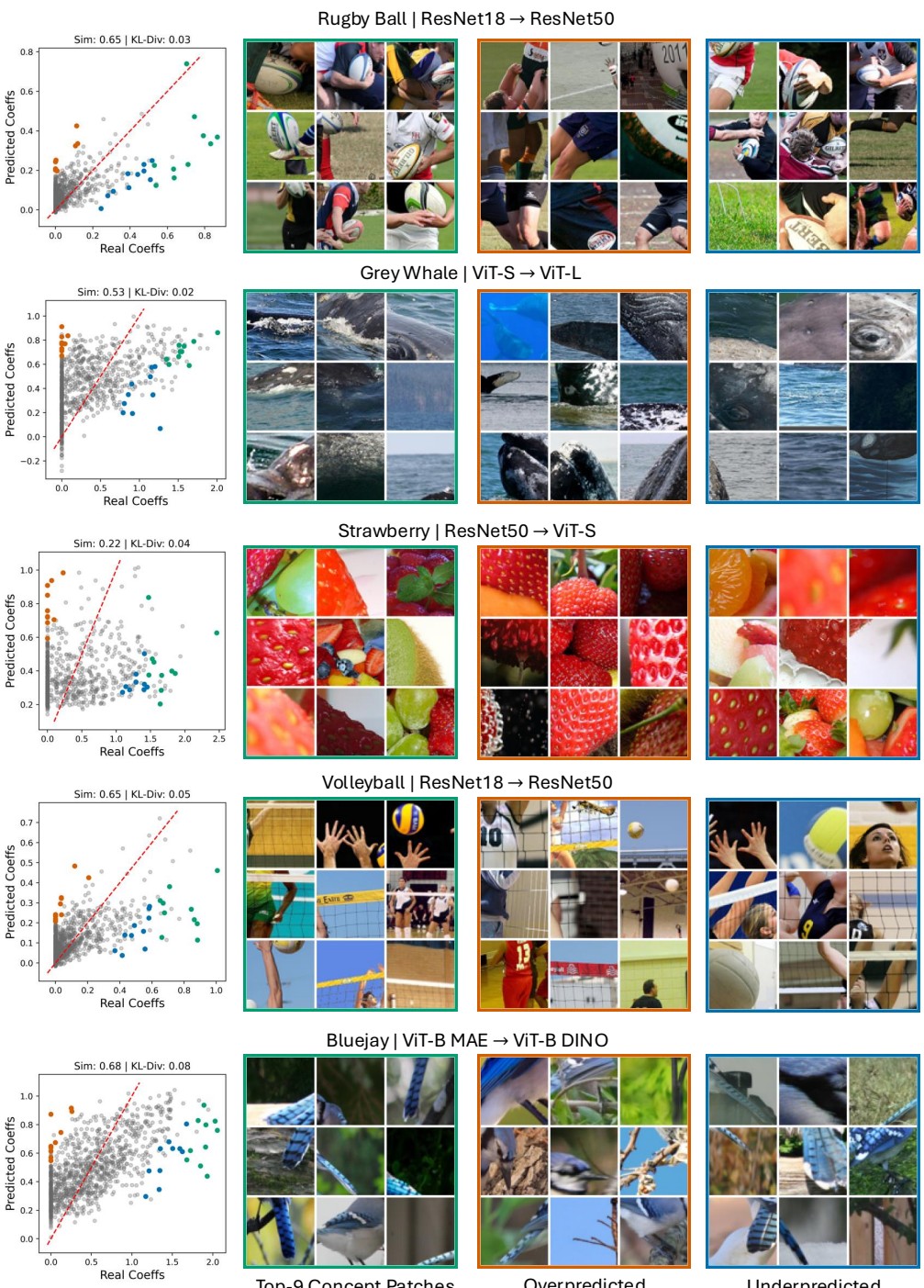

Figure A3: **Qualitative Samples.** In each row, we show visualizations for selected concepts from different model comparisons. In the first column of each row, we show scatter plots between real and predicted concept coefficients. Colored points mark the top-9 images in different subregions of the scatterplot. Each subregion indicates a different aspect of dissimilarity. (Green): Top-9 images for the real concept. These images are used to help the user understand what the target concept pays attention to. (Orange): Top-9 images that are over-predicted by the contrasted model. You may need to zoom in to best analyze the image grids. (Blue): Top-9 images that are underpredicted by the contrasted model. We discuss possible interpretations of the concepts in Appendix A.2. See Sec. 3.4 for a detailed breakdown of how to interpret these plots.

**Volleyball | RN18 → RN50**

The RN50 has learned to encode volleyball players in a variety of active positions. There seems to be an emphasis on hands and arms near or above the net. The RN18 under-predicts close-up images of volleyballs and players at the net and it tends to over-predict images containing balls high in the air and close-ups of nets/grids. This suggests that the regression model is trying to reproduce the behavior of the RN50 concept using a variety of related features learned by the RN18. We explore the top-6 neurons that most contribute to the regression model's prediction of this concept in Fig. A6. We compute the neuron contribution using permutation feature importance (Appendix B). We find that the regression model uses RN18 neurons that are highly sensitive to volleyballs in the air, players near nets, and close-ups of nets qualitatively explaining some of the differences between the predicted and real concept.

**Bluejay | ViT-B MAE → ViT-B DINO**

The DINO model has isolated the tail of the bluejay as an important visual feature. The regression model under-predicts images with bluejay tails and over-predicts random images, suggesting that this concept is not independently encoded in the feature space of the MAE.

### A.3 LLVM Generated Concept Difference Analysis

In Sec. 4.5, we provide an example that uses an LLVM for analyzing the image collages produced by RSVC. In Fig. A4, we describe the procedure in detail. We apply this method to the concept comparisonsin Fig. A3 and report the results below. We find that the LLVM annotations are fairly similar to the manual annotations.

**Rugby Ball | RN18 → RN50**

**IC1**: A collage featuring various close-ups of players holding rugby balls during gameplay, emphasizing hands and ball contact.
**IC2**: A collage with mixed shots of rugby gameplay, focusing on players' lower bodies, shorts, and action involving the rugby ball from different perspectives.
**Similarity**: Both collages depict scenes from rugby games and include elements of players and rugby balls in action.
**Difference**: IC1 focuses on hand interactions with the rugby ball, while IC2 emphasizes lower body dynamics and broader gameplay elements.
**Semantically different**: [Yes] IC1 highlights ball handling, whereas IC2 portrays physical movement and positioning.

**Grey Whale | ViT-S → ViT-L**

**IC1**: A collage featuring close-up views of whales, focusing on surface water interaction and detailed textures of their bodies.
**IC2**: A collage showing whales from various perspectives, including body parts like tails, heads, and fins, with more diverse settings and details.
**Similarity**: Both collages depict whales and highlight different parts of their bodies in aquatic environments.
**Difference**: IC1 emphasizes close-up textures and water interaction, while IC2 provides broader views of whales, showing distinct body parts and their environment.
**Semantically different**: [Yes] IC1 focuses on textures and surface detail, while IC2 captures a wider perspective of whale features and behaviors.

**Strawberry | RN50 → ViT-S**

**IC1**: A collage featuring various fruits, with a focus on strawberries along with other fruits like grapes, blueberries, and kiwi, emphasizing vibrant colors and diversity.
**IC2**: A collage entirely centered on strawberries, showing detailed close-ups of their textures and seeds in different lighting and angles.
**Similarity**: Both collages prominently showcase strawberries and their textures.
**Difference**: IC1 includes a mix of other fruits alongside strawberries, while IC2 exclusively focuses on strawberries and their intricate details.
**Semantically different**: [Yes] IC1 presents a broader context of mixed fruits, whereas IC2 isolates strawberries as the sole subject.

**System Prompt**

I am going to ask you to analyze image collages. You will receive two image collages and your task is to: (1) provide one sentence to describe each image collage. (2) Provide one sentence to describe similarities between the image collages and one sentence to describe the differences between the image collages. (3) Make a judgement on whether the two collages convey different semantic information and summarize it in a few words. The format of your output should be organized as follows:

IC1: desc
IC2: desc

Similarity: desc
Difference: desc

Semantically different: (Boolean Answer) - desc

**Input**

**Output**

**Analysis**

**IC1**: A collage featuring various close-ups of players holding rugby balls during gameplay, emphasizing hands and ball contact.

**IC2**: A collage with mixed shots of rugby gameplay, focusing on players' lower bodies, shorts, and action involving the rugby ball from different perspectives.

**Similarity**: Both collages depict scenes from rugby games and include elements of players and rugby balls in action.

**Difference**: IC1 focuses on hand interactions with the rugby ball, while IC2 emphasizes lower body dynamics and broader gameplay elements.

**Semantically different**: **Yes** - IC1 highlights ball handling, whereas IC2 portrays physical movement and positioning.

Figure A4: **LLVM Concept Difference Analysis.** We use ChatGPT-4o (Achiam et al., 2023) to analyze concept differences. We provide a general system prompt asking the LLVM to describe the collages, provide a description of the similarities and differences between the collages, and provide a final judgement on whether there is a semantic difference between the collages. We provide an example for the rugby ball class. The LLVM receives the image collage (IC) corresponding the top-k concept images (IC1) and the over-predicted concept images (IC2), see Fig. 5 for more details on image collages. Here, the LLVM notices that IC1 focuses on hands and ball contact, whereas the IC2 focuses more on lower bodies. The described difference matches the manual annotation provided in Appendix A.2.

**Volleyball | RN18 → RN50**

**IC1**: A collage depicting volleyball gameplay, focusing on players' hands, the volleyball, and interactions around the net during action-packed moments.
**IC2**: A collage showing volleyball gameplay as well but with a wider variety of scenes, including players near the net, different environments, and more casual setups.
**Similarity**: Both collages revolve around volleyball and the central theme of gameplay near the net.
**Difference**: IC1 emphasizes action shots with players and the ball in dynamic moments, while IC2 includes a mix of gameplay contexts and less action-oriented scenes.
**Semantically different**: [Yes] IC1 focuses on the intensity and dynamics of volleyball action, whereas IC2 portrays a broader variety of volleyball scenarios and contexts.

**Bluejay | ViT-B MAE → ViT-B DINO**

**IC1**: A collage of close-up views of blue jay feathers, primarily showcasing the distinctive blue and black patterns on their wings and tails.

**IC2**: A collage of blue jays in various settings, including full-body shots, close-ups of their heads, and natural habitats like branches and foliage.

**Similarity**: Both collages feature blue jays and focus on their unique physical features and natural environment.

**Difference**: IC1 highlights detailed patterns on blue jay feathers, while IC2 captures a broader context of the bird's appearance and behavior in its habitat.

**Semantically different**: [Yes] IC1 emphasizes feather patterns, whereas IC2 provides a holistic view of blue jays in their environment.

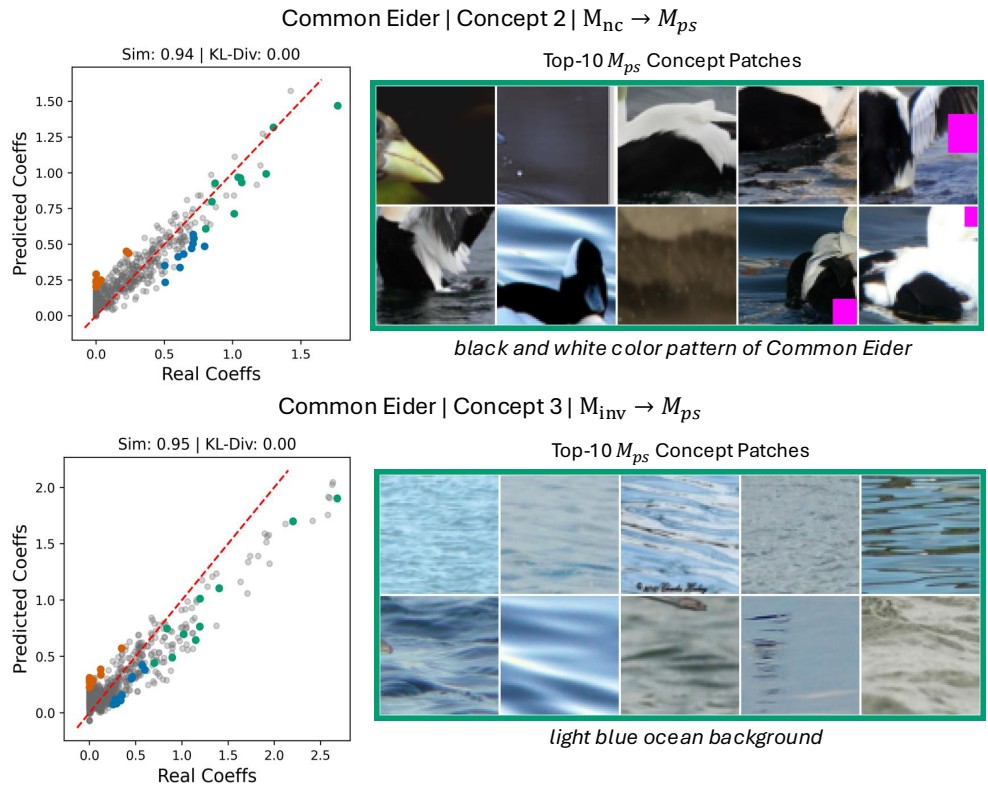

Figure A5: **Specificity of Toy Concept.** In Fig. 2, we showed that $M_{nc}$ is not able to predict the pink square concept from $M_{ps}$. In this figure, we show that the toy concept does not impact the similarity between other concepts learned by the networks. We visualize the top-10 patches from Concept 2 and Concept 3 of $M_{ps}$ in the same class (Common Eider). These concepts correspond to the white and black color pattern of the bird and a water background. Note that these models have been trained from scratch on NABirds resulting in a relatively low 34% accuracy. This leads to noisier concepts that are more challenging to interpret. Importantly, we can see that $M_{nc}$ still has a very high similarity score for these two concepts, highlighting the specificity of RSVC.

## A.4 SPECIFICITY OF RSVC ON THE TOY CONCEPT EXPERIMENT

In Sec. 4.1 we showed how RSVC can be used to distinguish between a model trained to use the pink square concept and a model trained to ignore the pink square concept. However, training with the pink square concept could have undesirable effects on other model concepts for the Common Eider class. In Fig. A5, we show two shared concepts that are unaffected by the pink square. This experiment suggests that our toy concept training procedure does not impact other concepts learned by the networks.

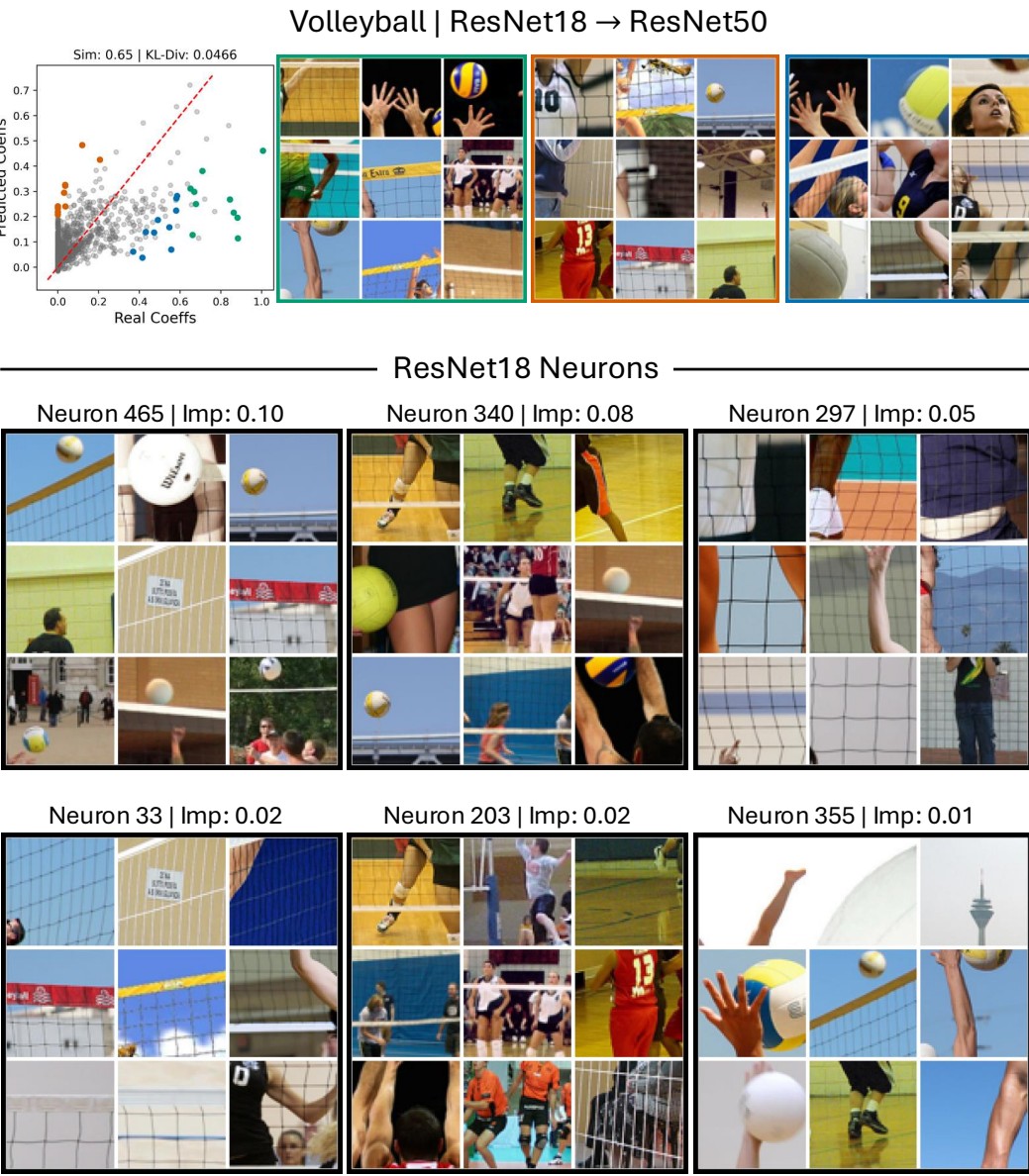

Figure A6: **Neuron Analysis For Volleyball Concept Difference** In Fig. A3 we visualized a RN50 concept for the volleyball class that the RN18 did not contain. In this figure, we explore the top-6 neurons used by the regression model to predict the RN50 concept. We find that Neuron 465 is sensitive to edges between a volleyball net and the background. It seems to mistake some grid-like textures for nets as well (image [1, 0], [1, 1], and [2, 0]). In addition, it seems to be sensitive to volleyballs high in the air. Neuron 340 seems to activate for athletes in indoor gyms and seems partial to lower bodies. Neuron 297 is sensitive to close-ups of nets with hands or arms in the frame. In summary, these neuron visualizations help to explain some of the images over-predicted by the regression model.

## A.5    VARYING RESNET-18 TRAINING

Next, we conduct controlled variations of model training and measure how it effects model concept similarity in the last layer. We train a ResNet-18 model on variations of the NABirds dataset.

We start by comparing two ResNet-18 models trained on NABirds with different seeds, $4834586$ (R18 s483) and $87363356$ (R18 s873). We find that, despite varying the seed during training, both

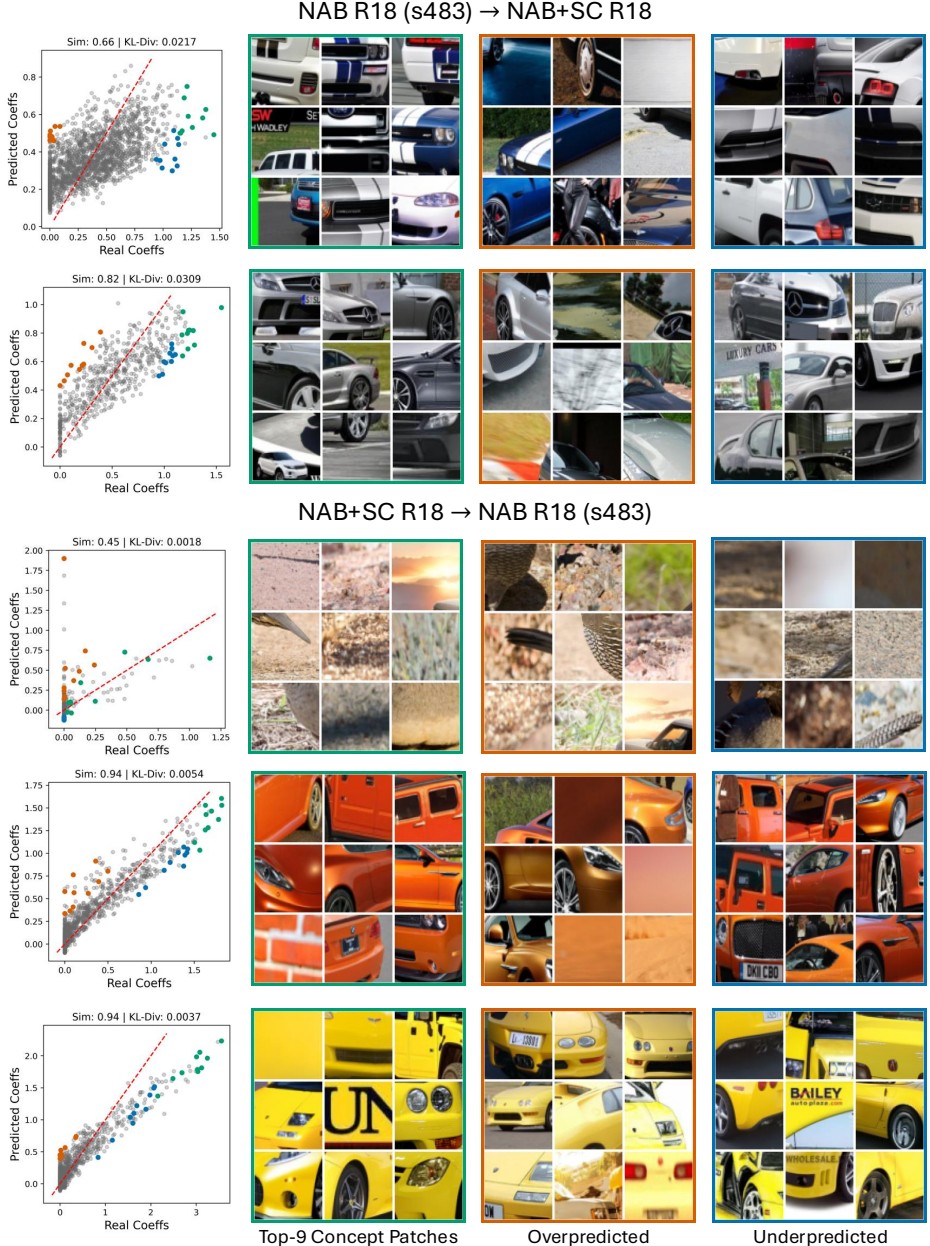

Figure A7: **Impact of Training on Stanford Cars.** In each row, we show visualizations for selected concepts from comparing R18 NAB+SC to R18 s483. In the first two rows, we visualize two R18 NAB+SC concepts that R18 s483 cannot reproduce. The first concept is a racing stripe that is associated with the Shelby Mustang. The R18 s483 model appears to sometimes entangle this concept with a blue color, irrespective of the car model. The second concept appears to be common features associated with Mercedes cars. For this concept, the difference between the two models is more abstract and challenging to interpret. We visualize NAB R18 s483 concepts in the next three rows. First, we show a R18 s483 concept that R18 NAB+SC is unable to predict. We see that this concept is very abstract without a clear pattern, but is generally related to sandy textures. Next, we visualize two car-related concepts from R18 s483. We find that these concepts are sensitive to the combination of the presence of a car and a specific color. For the orange car concept, the R18 NAB+SC makes small over-predictions with different shades of orange. For the yellow car concept, the over-predicted group shows a different shade of yellow and a specific style of car. A discussion of these results is available in Appendix A.5.

models discover highly similar concepts (Fig. A8**A**). Then, we train a ResNet-18 model on a modified version of the NABirds dataset in which waterbirds (169 classes) have been excluded during training (R18 NAB-WB). After training, the backbone is frozen and just the classification head is trained on the full dataset. We compare this model to R18 s483 trained on the full dataset (34%). We find that training without waterbirds results in a significant decrease in performance (25%) and, surprisingly, only a slight increase in dissimilar concepts (Fig. A8**B**).

We then explore if introducing novel features from an out-of-domain dataset would result in more dissimilar concepts. In this experiment, we train one model (R18 NAB+SC) on a combined dataset of NABirds and Stanford Cars (Krause et al., 2013) achieving 37% accuracy on the combined classification task. To compare to R18 s483, we freeze the backbone and re-train the model head on both NABirds and Stanford Cars, achieving 26% accuracy. We find training the backbone on Stanford Cars significantly increases concept dissimilarity (Fig. A8**C**). Interestingly, we find that the increase in dissimilarity is bi-directional, both models are less able to predict the concepts of their contrasted pair. In order to better understand the bi-directional nature of this dissimilarity, we visualize a few concepts from each model in Figure A7. These concepts were selected by (1) filtering concepts above the 75th percentile in delta KL-divergance, (2) visualizing the 15 lowest delta Pearson concepts and (3) manually selecting concepts that were easiest to interpret. We find dissimilar concepts from the R18 NAB+SC model that seem to be semantic concepts specific to car models. In contrast, we found no dissimilar car related concepts that met the criteria from R18 s483. Instead, we find that concepts that meet this criteria are from NABirds classes and tend to be challenging to interpret. We then visualize R18 s483 concepts that result in the largest kl-divergence when replaced by the predictions from R18 NAB+SC. When visualizing these concepts we find two car related concepts that seem to be primarily driven by color, but not car model. These results match the intuition that R18 NAB+SC should contain more complex, car model aligned concepts that can be used to better classify images from Stanford Cars. However, it is not a complete explanation of model behavior and further analysis is needed to make concrete statements about concept dissimilarity.

Finally, we compare a ResNet-18 trained on ImageNet and fine-tuned on NABirds to a NABirds model (R18 ImgNet PT). Unsurprisingly, this results in the most dissimilar concepts (Fig. A8**D**). In sum, we find that larger changes during training results in more dissimilar concepts.

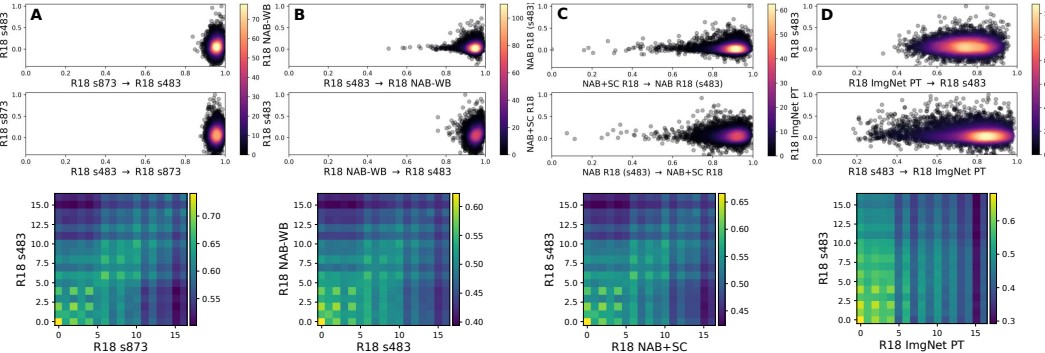

Figure A8: **Effect of Seed and Dataset on ResNet-18 Similarity.** We compare several pairs of ResNet-18 models while varying their training protocols. We use the same base model in all comparisons, a ResNet-18 model trained with the seed set to 4834586 (R18 s483). (A) We compare the base model to a model trained with seed 87363356 (R18 s873) and find that the two models are highly similar despite the change in seed. (B) We train a ResNet-18 on a modified dataset where we exclude 169 classes that belong to the coarse category of waterbirds (R18 NAB-WB). When comparing to the seed variation experiment, we see a slight increase in the number of dissimilar concepts. (C) We train a ResNet-18 on a combined dataset of NABirds and Stanford Cars (R18 NAB+SC). To compare to the base model, we freeze the base model's backbone and re-train the linear classifier on this combined dataset. We find that introducing Stanford Cars results in a significant increase in dissimilar concepts. (Right) Finally, we compare to a model pre-trained on ImageNet and fine-tuned on NABirds (R18 ImgNet PT). We find that training on ImageNet introduces many novel concepts that are dissimilar to the features of the base model.

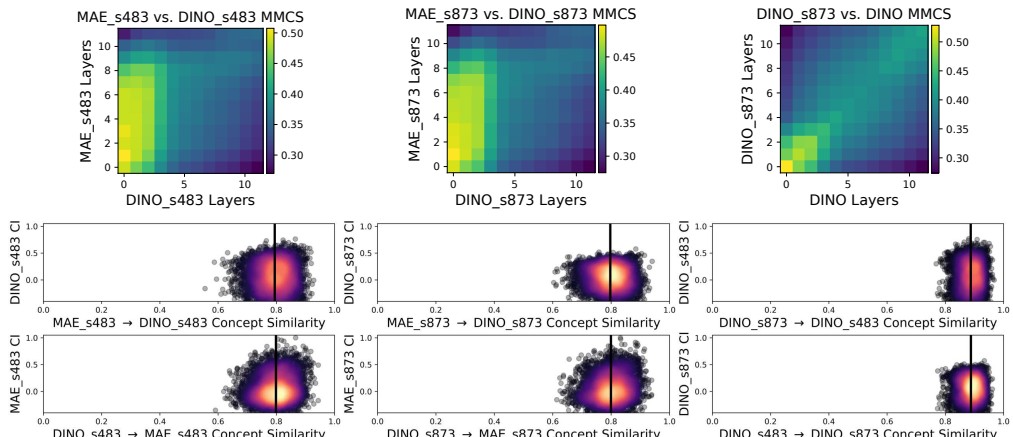

Figure A9: **DINO and MAE Seed Variation.** We explore the effects of varying seed on finetuning a DINO and MAE model on the NABirds dataset. **(Left):** We show layerwise and last layer comparisons of MAE_s483 vs. DINO_s483. These plots are reproductions from the main text. The black line denotes the average concept similarity. For this comparison, the average similarity in both directions is 0.80. **(Center):** We compare DINO_s873 vs. MAE_s873. We see a similar layerwise matrix and last layer similarity to DINO_s483 vs. MAE_s483. The average similarity for both models is, once again, 0.80. **(Right):** We compare DINO_s483 vs. DINO_s873 and find that there is a better layer-to-layer mapping in the layerwise comparison matrix. In addition, the average similarity in both directions is 0.89, higher than comparisons across the different pretraining strategies. Taken together, these results indicate that individual concepts change due to different seeds, but the global structure of the relationship between these models is not affected by seed.

## A.6 DINO AND MAE SEED VARIATION EXPERIMENTS

In Section 4, we compared a DINO pretrained model and a MAE pretrained model that were finetuned on the NABirds dataset. In those experiments, models were finetuned with the seed set to 4834586. In this section, we explore comparisons to models finetuned with a different seed, 87363356. The models finetuned with the new seed are denoted as DINO_s873 and MAE_s873. We compare two pairs of models, DINO_s873 to MAE_s873 and DINO_s483 to DINO_s873. In Figure A9, we show that changing the seed does change the concepts learned, but that the general relationship between the different pretraining strategies is preserved. In particular, we find that comparing DINO models finetuned with different seeds results in a higher average similarity (∼0.89) than models with different pretraining strategies (∼0.80), indicating that the seed has a smaller impact on finetuning than the initialization.

## A.7 SAME MODEL CONCEPT SIMILARITY VS. CONCEPT IMPORTANCE

In this section, we validate the feasibility of the regression task. Due to the reconstruction error inherent in decomposition methods, it is not possible to perfectly predict the concept coefficients from the activation matrix. However, in Fig. A10, we show that the regression models do well when trying to do same-model concept regression and significantly better than cross-model concept regression.

## A.8 ADDITIONAL REPLACEMENT TESTS

In Figs. A11 and A12 we visualize the replacement tests for the three pairs of models not presented in the main text. We see the same effects as before, with a decrease in similarity corresponding to increasing changes in model behavior. Notably, although DINO and MAE models (finetuned on NABirds) have high similarity relative to the other models, they show stronger changes in model behavior for smaller changes in similarity. However, it is not clear whether these differences are due to changes in dataset or changes in pretraining.

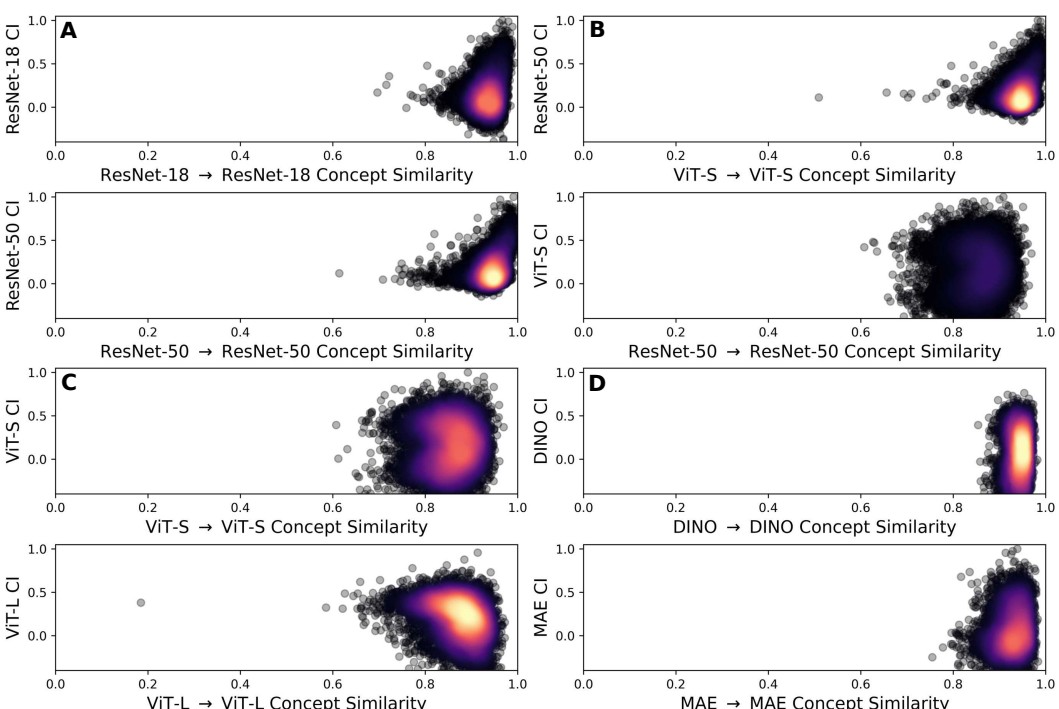

Figure A10: **SMCS vs CI.** We visualize same-model concept similarity (SMCS) against the concept importance. We find that reconstructing more important concepts tends to be easier for ResNets. However, for some ViT models, there can be important learned concepts that are hard to predict. Importantly, SMCS is significantly higher than CMCS indicating that the regression task is feasible.

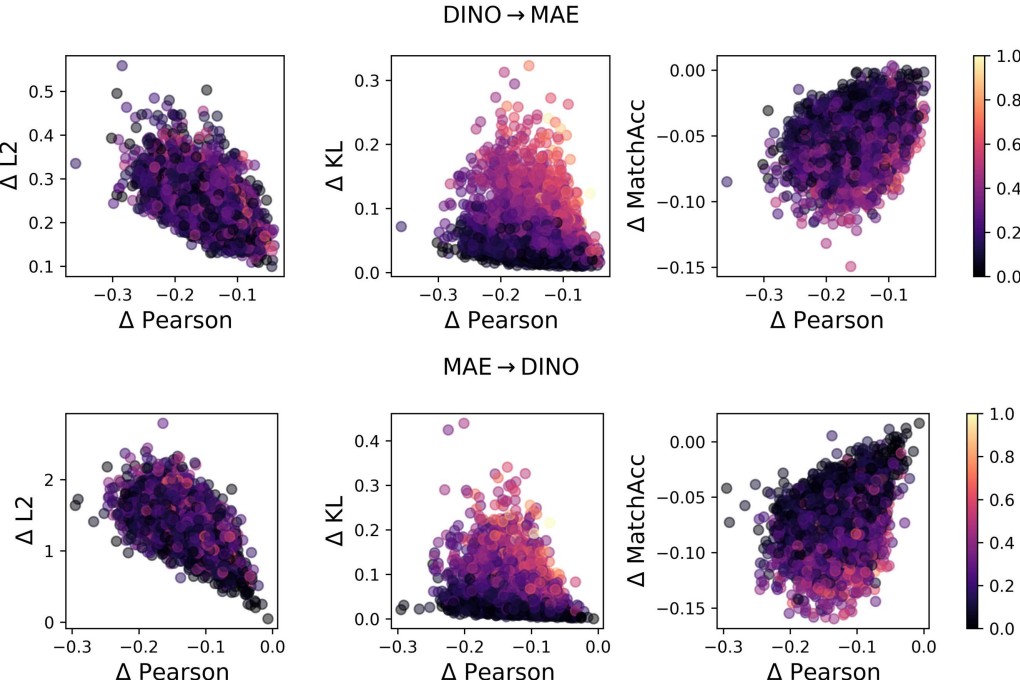

Figure A11: **Replacement Test for DINO vs. MAE (NABirds).** We find that for the DINO vs. MAE comparison. As Pearson correlation decreases the $l_2$-distance increases, KL-divergence increases, and the match accuracy decreases. Notably, the Pearson correlation decreases a smaller amount than for the other three pairs of models, but the change in the three metrics is on the same order as the other comparisons. This suggests that these two models are more sensitive to changes in a concept.

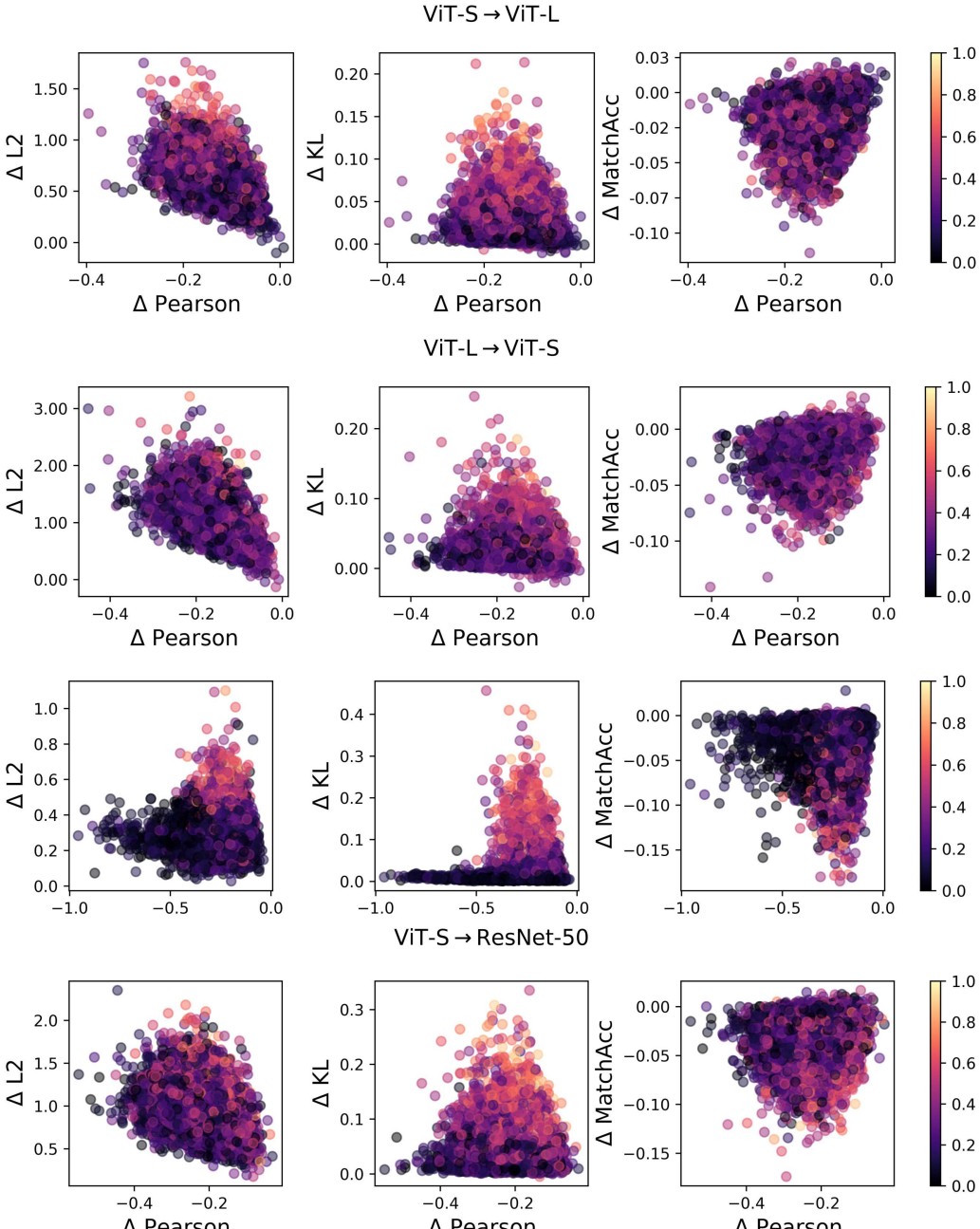

Figure A12: **Replacement Test for ViT-S vs. ViT-L and RN50 vs. ViT-S.** We find that for these model comparisons, as Pearson correlation decreases the $l_2$-distance increases, KL-divergence increases, and the match accuracy decreases.

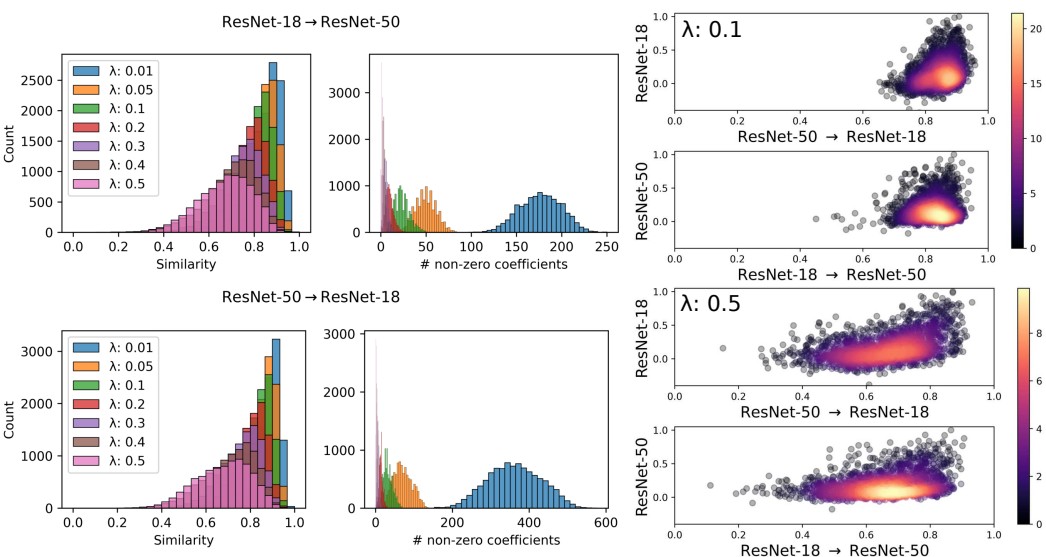

Figure A13: **Impact of Regularization on Regression.** Here we vary the $\lambda$ for the $l_1$ penalty on the regression model. We use the first 200 classes of ImageNet for these visualizations. In the left column, we visualize the distribution of similarity values for each value of $\lambda$. In the center, we visualize the number of non-zero coefficients. In the right column, we visualize the similarity vs. importance plots for $\lambda = 0.1$ and $\lambda = 0.5$. We find that, as expected, increasing the $l_1$ penalty reduces similarity by increasing the number of zeroed coefficients. In all experiments in the paper, we use an $l_1$ penalty of $0.1$.

## B  ADDITIONAL IMPLEMENTATION DETAILS

### B.1  CONCEPT EXTRACTION

For each model considered in this study, we provide information about concept extraction in Tab. A2. First, images are resized to 224x224 and then processed into 16 evenly spaced 64x64 pixel patches. Patches are then resized back to the image resolution of the network. We sample 100 images per model for concept extraction. All models taken from the timm library were trained with Inception style random cropping. Custom trained models were trained using random resized cropping with horizontal flipping. This ensures that the resized patches are in-domain for the network. To produce an activation matrix $\mathbf{A}$ that can be decomposed, the outputs of the network are processed. The ResNets (He et al., 2016) produce outputs with a batch $b$, channel $c$, height $h$ and width $w$ dimension. To create a matrix that can be decomposed, we use global average pooling over the $h$ and $w$ dimensions. The ViTs (Dosovitskiy, 2021) produce outputs with a batch $b$, sequence $s$, and feature dimension $d$. We select the class token from the sequence dimension resulting in a two-dimensional matrix. We use NNMF for ResNets since they contain ReLU layers and can produce positive only activations. NNMF restricts the $\mathbf{U}$ and $\mathbf{W}$ matrix to be positive. For, ViT models, we use Semi-NMF (Ding et al., 2008) which allows for both positive and negative values in the $\mathbf{W}$ matrix, but requires positive values in the $\mathbf{U}$ matrix. We use a non-negative least squares solver to fit coefficients to a new set of data points:

$$\min_{\mathbf{U}_1} \quad \|\mathbf{A}_1 - \mathbf{U}_1\mathbf{W}_1\|_2^2,$$
$$\text{subject to} \quad \mathbf{U}_1 \geq 0. \tag{5}$$

### B.1.1  CONCEPT INTEGRATED GRADIENTS

Integrated gradients measures the importance of each pixel by averaging the gradients of the input image, as the input image is varied from a baseline value to its true value (Sundararajan et al., 2017). To compute concept integrated gradients the formulation is modified. Let $h_1$ represent the head of $M_1$, i.e., the final layer(s), and $\mathbf{A}_1$ be the output activations from the layer preceding $h_1$. As

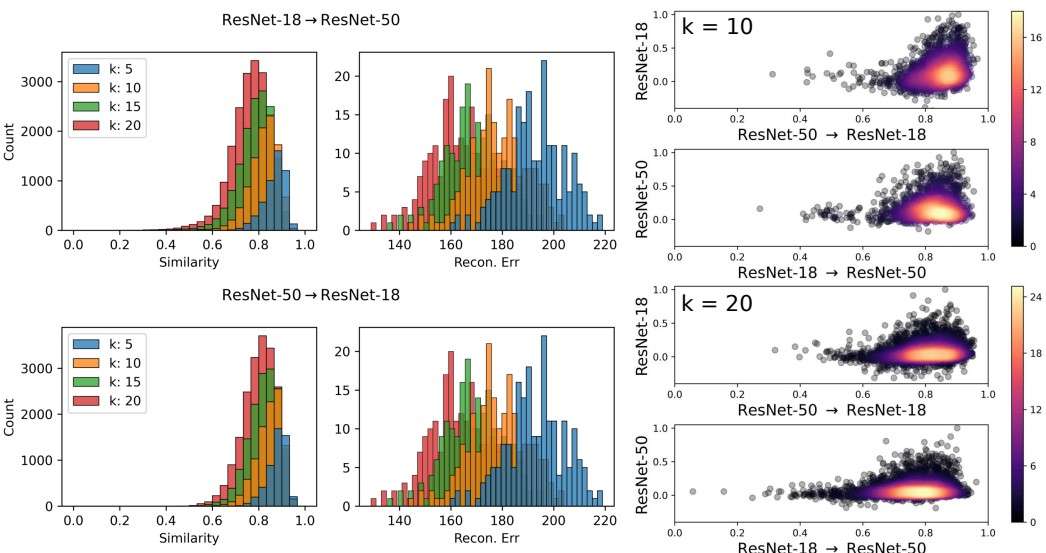

Figure A14: **Impact of Number of Concepts on Similarity.** Here we vary $k$, the number of concepts in the dictionary and explore the impact on the similarity distribution. We use the first 200 classes of ImageNet for these visualizations. In the left column, we plot the distribution of similarity scores for 5, 10, 15, and 20 concepts. In the center column, we visualize the distribution of reconstruction errors for different number of concepts. As expected, increasing the number of concepts results in lower reconstruction errors. In the right column, we visualize similarity vs. importance for 10 and 20 concepts. We observe that increasing the number of concepts disproportionately increases the number of dissimilar concepts. For all results in the paper we use 10 concepts.

described earlier, we factorize $\mathbf{A}_1 \approx \mathbf{U}_1\mathbf{W}_1$. We denote row vectors of $\mathbf{U}_1$ as $\mathbf{r}_1^i \in \text{Rows}(\mathbf{U}_1)$, such that $\mathbf{r}_1^i \in \mathbb{R}^{1 \times d}$. To link model predictions to learned concepts, we compute model predictions as

$$\hat{\mathbf{z}}_1^i = h_1(\mathbf{r}_1^i \mathbf{W}_1), \tag{6}$$

where $\hat{\mathbf{z}}_1^i \in \mathbb{R}^{1 \times d}$ is a row vector of prediction probabilities. Then, to compute concept integrated gradients, we average over the gradients as we linearly step from a baseline vector $\mathbf{r}^b = 0$ to $\mathbf{r}_1^i$

$$\varphi(\mathbf{r}_1^i) = (\mathbf{r}_1^i - \mathbf{r}^b) \times \int_0^1 \nabla_{\mathbf{r}_1^i} h_1\left(\left(\alpha\mathbf{r}^b + (1-\alpha)(\mathbf{r}_1^i - \mathbf{r}^b)\right)\mathbf{W}\right)d\alpha. \tag{7}$$

Thus, for each class and concept we have a single value that represents the importance of that concept to model decisions. We implement concept integrated gradients based on the implementation in the xplique library (Fel et al., 2022a). For all experiments we integrate over 30 steps.

## B.2 CONCEPT SIMILARITY

**Correlation**. Pearson and Spearman are computed using scikit-learn (Pedregosa et al., 2011). We use 50 images for each class from the training set of the model. Images are resized to $224 \times 224$. We use a patch size of $64 \times 64$ resulting in 16 patches per image. Thus, Pearson and Spearman correlation is computed using 800 total patches per class. The patches are resized and passed through the model to generate activations at a given layer.

**Regression**. We use lasso-regression (Tibshirani, 1996) with a 0.1 weight on the $l_1$ penalty. We visualize the effect of this parameter on similarity in Fig. A13. Regression models are trained on the activations from at least 5 images (80 patches) and at most 200 images (3200 patches) sourced from the original training split of the dataset. For each concept and class, we train five lasso-regression models on different equally sized folds. The regression model weights are averaged and then the model is evaluated on images from the validation/test split of the original dataset. The inputs and targets are standardized to have a mean of zero and a standard deviation of one. The regression is trained using the Celer library (Massias et al., 2018). Finally, the predicted coefficients are unnormalized before the Pearson and Spearman correlation are computed. To compute feature importance

scores for regression models, we use the permutation feature importance implementation from scikit-learn (Pedregosa et al., 2011). We use the default parameters with 5 repeats and the random state set to 0.

**Layerwise Comparisons**. We list all of the layers used in the layerwise comparisons in Tab. A4.

**Spearman Correlation**. In all experiments, we found Spearman correlation to be very similar to Pearson correlation, thus we have excluded these results.

**Visualizing Dis-similar Concepts**. We select one patch per image in order of maximum concept coefficient. The top $n$ patches for the real images are excluded from the pool of images used to visualize the under-predicted and over-predicted coefficients.

### B.3 COMPUTATIONAL COST

All experiments were conducted using on a machine with an AMD Ryzen 7 3700X 8-Core Processor and a single GeForce RTX 4090 GPU. In Table A3, we detail the computational cost of each step of our proposed method. For a comparison between a ResNet-18 and a ResNet-50 on all 1000 classes of ImageNet, RSVC takes approximately 20 hours.

Table A1: **Model performance.**

| Model | timm Model | ImageNet Accuracy | NaBirds Accuracy |
|---|---|---|---|
| ResNet-18 | resnet18.a2_in1k | 70.6% | – |
| ResNet-50 | resnet50.a2_in1k | 79.8% | – |
| ViT-S | vit_small_patch16_224.augreg_in21k_ft_in1k | 81.3% | – |
| ViT-L | vit_large_patch16_224.augreg_in21k_ft_in1k | 85.8% | – |
| DINO ViT-B | vit_base_patch16_224.dino | – | 71.2% |
| MAE ViT-B | vit_base_patch16_224.mae | – | 71.2% |

Table A2: **Concept extraction.**

| Model | Layer Post-processing | Method | Number of Concepts | Patch Size | Recon. Error (Last Layer) |
|---|---|---|---|---|---|
| ResNet-18 | GAP | NNMF | 10 | 64 | 176.2 |
| ResNet-50 | GAP | NNMF | 10 | 64 | 205.5 |
| ViT-S | Class Token | Semi-NMF | 10 | 64 | 926.9 |
| ViT-L | Class Token | Semi-NMF | 10 | 64 | 1650.8 |
| DINO ViT-B | Class Token | Semi-NMF | 10 | 64 | 191.0 |
| MAE ViT-B | Class Token | Semi-NMF | 10 | 64 | 656.5 |

Table A3: **Computational cost for ResNet18 vs. ResNet-50 on ImageNet.**

| Step | sec/it | Total Time |
|---|---|---|
| Activation Extraction (RN50) | 1.50 | 25m |
| Concept Extraction (RN50) | 2.00 | 33m |
| Concept Comparison (CMCS) Last Layer | 9.00 | 2h30m |
| Concept Comparison (MCS) All Layers | 14.56 | 4h |
| Concept Int. Grad | 41.56 | 11h 30m |
| Regression Evaluation | 2.30 | 38m |
| Total Time | - | 19h36m |

Table A4: **Selected layers.**

| Model | Layers |
|---|---|
| ResNet-18 | act1, layer1.0.act1, layer1.0.act2, layer1.1.act1, layer1.1.act2, layer2.0.act1, layer2.0.act2, layer2.1.act1, layer2.1.act2, layer3.0.act1, layer3.0.act2, layer3.1.act1, layer3.1.act2, layer4.0.act1, layer4.0.act2, layer4.1.act1, layer4.1.act2 |
| ResNet-50 | act1, layer1.0.act1, layer1.0.act2, layer1.0.act3, layer1.1.act1, layer1.1.act2, layer1.1.act3, layer1.2.act1, layer1.2.act2, layer1.2.act3, layer2.0.act1, layer2.0.act2, layer2.0.act3, layer2.1.act1, layer2.1.act2, layer2.1.act3, layer2.2.act1, layer2.2.act2, layer2.2.act3, layer2.3.act1, layer2.3.act2, layer2.3.act3, layer3.0.act1, layer3.0.act2, layer3.0.act3, layer3.1.act1, layer3.1.act2, layer3.1.act3, layer3.2.act1, layer3.2.act2, layer3.2.act3, layer3.3.act1, layer3.3.act2, layer3.3.act3, layer3.4.act1, layer3.4.act2, layer3.4.act3, layer3.5.act1, layer3.5.act2, layer3.5.act3, layer4.0.act1, layer4.0.act2, layer4.0.act3, layer4.1.act1, layer4.1.act2, layer4.1.act3, layer4.2.act1, layer4.2.act2, layer4.2.act3 |
| ViT-S | block0, block1, block2, block3, block4, block5, block6, block7, block8, block9, block10, block11 |
| ViT-L | block0, block1, block2, block3, block4, block5, block6, block7, block8, block9, block10, block11, block12, block13, block14, block15, block16, block17, block18, block19, block20, block21, block22, block23 |
| DINO ViT-B | block0, block1, block2, block3, block4, block5, block6, block7, block8, block9, block10, block11 |
| MAE ViT-B | block0, block1, block2, block3, block4, block5, block6, block7, block8, block9, block10, block11 |

