# OpenReview forum: "Representational Similarity via Interpretable Visual Concepts"
_ICLR.cc/2025/Conference — ICLR 2025 Poster_

### Official Review · Reviewer_73oU · 2024-10-23

**Soundness:** 2
**Presentation:** 2
**Contribution:** 2
**Rating:** 6
**Confidence:** 3

**Summary:**

### Post rebuttal

---------------

The rebuttal clarifies most of my concerns. I think this work may provide some direction on how to compare models in terms of model explanations.

For the final version, I urge the authors to:

1. Point to some of the GPT based model explanations to Fig 2 and Section 3.4.
2. Clean the plots in Fig 3 and Fig 4 to remove the majority of less important points to make the trends more visible.
3. Add to the main paper, that they used a separate validation set to tune lambda from Eq. 4

---------

This work aims to analyze similarity between two trained networks in terms of what visual concepts they encode. It builds on top of prior work (Fel et al 2023) which provides a formulation to extract “concepts” from learned network. Namely, the activations of a network $A  \in R^{N \times D} $ are decomposed to form $U \in R^{N \times k}$ (concept coefficient matrix)  and $W  \in R^{k \times D}$ (concept basis). Given $U_1$, $R_1$, $U_2$ and $R_2$ of two different models, they propose “correlation” and “regression” to measure concept similarity. Correlation (MMCS) (loosely) measures the correlation between $U_1$ and $U_2$, while regression (CMCS) measures how well $U_2$ can be decoded from $U_1$ using a linear model (and vice versa).

The paper performs three experiments: comparing concept importance vs CMCS, how changes in concept similarity affect final outputs and layer-wise conceptual similarity using MMCS.

**Strengths:**

The paper explores a new area which is representational similarity using visual concepts. Finding interpretable ways on how different networks are similar may offer valuable insights to the community.

**Weaknesses:**

On reading the title, I was expecting insights that show what sort of visual concepts are learned by a model, and what is different as scale, data and architecture is varied. But I found these insights not present in the paper. The paper performs three sets of experiments but its a bit unclear why these are interesting to study. For example:

* In Figure 5, the authors investigate their proposed metrics layer-wise across various models. However, what additional insights does this offer over (Raghu et al 2021)?
* In Figure 4, high importance features show high correlation between L2 and pearson coefficients, But this is also a bit obvious.

I also found the presentation sometimes difficult to follow. See below for detailed feedback.

**Questions:**

* The abstract says that this work presents a practical tool to discover the cause of model failures but this discussion is missing in the main text.
* L167 - L169. Next, as visual features are highly correlated, patches from each image are sampled to form a set X c 1 of “concept proposal” images. More detail is required on how these patches are extracted from the image? Are they just randomly sampled, if yes then how many? And is each patch resized independently to the original image resolution?
* L202 - L203. It’s unclear why there is a tradeoff between computational cost and error between these two techniques.
* L201 - 211. $R^{\rho}$ and $R^{S}$ are mentioned, so which one does R employ?
* L214: Eq 2) takes the max across all columns for each row and vice-versa. IIUC, this is because there is no direct correspondence between the columns in U1 and the columns in U2. That is the first concept of U1 may represent something completely different from the first column U2. So to measure similarity, one matches each column of U1 to the column that has maximum correlation in U2. Even though this may be obvious to the authors, it will be nice if some discussion is added.
* L297: Eq1) How is $\lambda$ tuned?
* L348 - L349 For our exploration with DINO and MAE, we fine-tune the models on NABirds (Van Horn et al., 2015). Why do this?
* Fig 2) plots the correlation plots between $U_1 \in R ^{N \times K}$ and $U_{1\rightarrow 2}$. Are both matrices flattened to 1D to plot the correlation plot. What is the value of n and k?
* Fig 2) In both orange and light blue grids, there are grid-like textures and hand and arms at net. So, I find the descriptions and explanation unconvincing
* Fig 3). The paper introduces low, medium CI and concept similarities without any precise definitions. One could for example precisely define the top 5% percentile of concept importance as high and so on.
* Fig 3) How many points are in each subplot of Fig 3. Some amount of detail atleast for one subplot may be required.
* Fig 3) The paper says that “model differences are largely driven by medium similarity, medium importance concepts”. I found this conclusion hard to believe for the y-axis, the center of each heatmap seems to be close to 0.0 For the x-axis, it is indeed centered at medium importance.
* Fig 4) The presentation will be clearer if separate plots are presented for high importance and low importance plots.

---

> ### Author Response · Authors · 2024-11-23
> **73oU Response Part 1**
>
> Thank you for your review.
>
> > **[73oU-1]** Unclear motivation of work. Expected a more systematic exploration of changes in features as scale, data and architecture is varied.
>
> In this revision, we clarify the motivation of our work in the abstract, introduction, and related works. Our primary contributions are (1) defining the problem of visualizing model differences, (2) introducing a new method, RSVC, to address the problem, and (3) establishing that RSVC measures what we intend it to measure. We have updated the manuscript to clarify the importance of this problem for the field of XAI. We believe that by introducing this problem and our approach, the XAI community should have an exciting, new way to think about what makes a good explanation for models. We expect that future research will improve the method and apply it for deeper insights about model differences. We also make some initial steps towards applying it on more systematic variations during training and exploring some of the features that are discovered. In these new experiments (Sec. A.2, A.3, Figs. A.2-A.4 A.7) , we manipulate the training of a ResNet18 on NABirds and show how similarity changes. We find that increasing differences in training protocol results in progressively more dissimilar and important concepts. We explore one pair of models in these experiments and generate qualitative visualizations of feature differences. We detail these experiments in the response to reviewer **[Wky2-1]** .
>
> > **[73oU-2]** Why conduct the experiments in Figs 3, 4, and 5?
>
> One of the primary goals of this work is to establish the validity of the comparison method. We include these figures because they answer critical questions that assess the validity and usefulness of the proposed tool. Fig 3 shows that RSVC can discover concepts that are *unique* and *important* to a model, Fig 4 links representational differences in two model’s to specific changes in a model’s behavior, and Fig 5 (1) verifies existing observations in the literature (2) provides a different perspective into how layers relate in terms of concepts. As a small step in trying to gain deeper insights, we provide a new experiment in the Appendix A3 comparing a model trained on NABirds to a model trained on NABirds *and* Stanford Cars. We find that there are some consistent differences in the types of features learned. The model trained on both NABirds and Stanford Cars exhibits complex concepts that are semantically aligned with specific car models. For example, one discovered concept is sensitive to the racing stripe on a type of muscle car. In contrast, car concepts for the model trained solely on NABirds appear to be sensitive to a particular color and style of car, but are not specific to the car model. RSCV opens the door to future focused evaluation on a specific model and a small set of classes to present a complete picture of model behavior. In this work we have focused on a broader exploration of the tool's properties to demonstrate its potential.
>
>
> > **[73oU-2.1]** What does Fig 5 demonstrate?
>
> Fig 5 provides value in a few ways: (1) it shows a subtle, but interesting phenomena suggesting that concept similarity is lower in middle layers than in the final layers, (2) acts as an important sanity check on our method and (3) provides insights into layerwise similarity when comparing DINO and MAE pre-trained models. Points (1) and (3) are new results that were not explored in Raghu et. al. 2021.
>
> > **[73oU-2.1]** Is the L2 replacement test in Fig 4 obvious?
>
> The replacement test helps assess whether prediction error for a regression model is meaningful, since error could be localized to images in which the concept contributes very little to model activations. While the result for L2 is expected, the main point of including this figure is to look at how it changes the logits (kl-div) and match accuracy. In the figure, we include L2 for completeness.
>
> > **[73oU-3.1]** Abstract states it is a practical tool for model failures, but not discussed in the paper.
>
> We have modified the language in the abstract and introduction. Our method provides hypotheses for differences in model behavior, which paves the way for future work to provide a complete explanation for model failures.
>
> > **[73oU-3.2]** Implementation details.
>
> We expand on implementation details in the Appendix (Sec. B). Images are resized to 224x224 and patched into 64x64 patches (evenly spaced), resulting in 16 patches per image. Each patch is resized back to the original image resolutions. Models are always trained with either inception style cropping or random resized cropping, ensuring that patches are in-domain for the network.

---

> ### Author Response · Authors · 2024-11-23
> **73oU Response Part 2**
>
> > **[73oU-3.3]** Computational cost trade-off.
>
> For a given class and layer, comparing two models using MCS only requires using Pearson correlation whereas using RSVC requires training five lasso regression models. We give more details about computational cost in Table A.3 and Section B.
>
> >  **[73oU-3.4]** Pearson and Spearman
>
> While we compute plots for both Pearson and Spearman correlation, we find little difference with the two methods. Therefore, all plots use Pearson correlation. We will include a comparison in the appendix in the final version.
>
> > **[73oU-3.5]** Clarification of motivation for maximum in MMCS.
>
> Your understanding of the reason for the maximum is correct. We add some text clarifying this point in the updated version of the manuscript (Line 214-216).
>
> >  **[73oU-3.6]**  How is lambda selected?
>
> We have conducted an ablation of the L1 penalty in Fig. A12 of the appendix. We find that the histogram of similarity values is consistent for lamba values 0.01, 0.05, and 0.1, but that similarity values drop significantly when going to 0.5. In our work, we choose 0.1.
>
> >  **[73oU-3.7]** Why NABirds for DINO and MAE.
>
> We wanted to explore a second dataset (NABirds) that had less images available for training the regression model. We also include several ablation experiments (Secs. A.3, A.4; Figs. A4, A5, A7) on the NABirds which are feasible due to the smaller size of the dataset.
>
> >  **[73oU-3.8]** How are correlation plots generated?
>
> The correlation plots are for a single, specific concept (a column in $U_1$). We indicate this on the x and y labels with a vector $u^a_1$ and $u^a_{2\rightarrow 1}$ where $a$ represents an arbitrary column vector from $U$ and it is also noted in Line 079.
>
> > **[73oU-3.9]**  Differences in Fig. 2 are not clear.
>
> Some reviewers found Fig. 2 to be challenging to interpret. We swap it with a sample from the barbell class that has much clearer differences. There are more examples provided in the appendix. We also conducted a new experiment using ChatGPT (Sec. A5, Fig. A6) to provide interpretations given two image collages. We ask it to provide an overview of both similarities and differences and find that it roughly matches the manual annotations we provide.
>
> > **[73oU-3.10]** Defining/grouping low, medium and high importance.
>
> in Section A.3 and Figs. A4, A7, we have added an experiment exploring the impact of training with and without extra data on the features learned by the network. In these experiments, we have detailed a more systematic approach to surfacing interesting low similarity features.
>
> > **[73oU-3.11]**  Number of points in Fig. 3
>
> There are 10 concepts per class and 1000 classes for the ImageNet experiments, resulting in 10000 points. There are 555 classes and 10 concepts per class for the NABirds experiments, resulting in 5550 points. We have also added color bars indicating the number of points vs. color intensity.
>
> > **[73oU-3.12]** Interpretation of Fig. 3 (medium importance vs zero/low importance).
>
> The x-axis represents similarity and the y-axis represents importance. As you have noticed, density along the x-axis peaks around medium similarity. On the y-axis, while there are many zero/low importance concepts it is clear in Fig. 4 (replacement test) that these concepts do not impact model behavior. Therefore, we believe it is fair to see that medium importance concepts drive the bulk of model behavior.
>
> > **[73oU-3.13]** Separating Fig. 4 into subplots.
>
> We have not yet created this plot, but will include it in the final version.

---

> > ### Author Response · Authors · 2024-11-27
> > **73oU Response Part 3**
> >
> > > **[73oU-3.6]** How is lambda selected?
> >
> > We have improved the L1 parameter sweep by increasing the granularity (Fig. A12). We see that, as expected, increasing the L1 penalty increases the number of zero coefficients. This results in lower similarity scores as L1 penalty. In this main paper, we have used an L1 penalty of 0.1. We also evaluate and visualize the impact of different numbers of concepts (Fig. A13). We see that increasing the number of concepts reduces the reconstruction error, but also results in more low/medium similarity concepts. We use 10 concepts for experiments in the main paper.
> >
> > > **[73oU-3.9]**  Differences in Fig. 2 are not clear.
> >
> > As indicated before, we have swapped Fig. 2 for a simpler example.
> >
> > In this new revision, we include a deeper analysis of the volleyball concept. The RN50 contains a concept for the volleyball class that has low similarity with the RN18. We explore the weights of the regression model trained on the RN18 activations to see which neurons were used to try and reproduce the RN50 concept. We compute the permutation feature importance for the neurons to identify the top 6 most important neurons. We then visualize the image patches that most activated these neurons (see Fig. A14). We find that neurons are sensitive to features like the boundary between volleyball nets and background, volleyballs in the sky, players near volleyball nets with a focus on lower bodies, and close-up pictures of nets with hands and arms. We also note that the most important neuron seems to activate for objects that appear to be **like** the boundary between a volleyball net and the background, but are in reality not a volleyball net at all. For example, the neuron activates strongly for a grid texture on a wall. When compared to the over-predicted images, the contributing neurons seem to qualitatively explain the increased sensitivity to certain types of images.

---

> > > ### Comment · Reviewer_73oU · 2024-11-29
> > > **Follow-up Question**
> > >
> > > Thanks for the rebuttal and additional clarifications. The added GPT experiments on labeling the "underpredicted" and "over predicted" patches are more convincing.
> > >
> > > I have a follow-up question. The authors explain mean-L2, mean-KL and match accuracy from L322-L323. Can the authors define (∆Pearson) precisely? It's the difference between which two pearson coefficients?

---

> > > > ### Author Response · Authors · 2024-11-29
> > > > **$\Delta$ Pearson Clarification**
> > > >
> > > > We compute two values, the same-model concept similarity (SMCS; $A_1 \rightarrow U1$) and the cross-model concept similarity (CMCS ; $A_2 \rightarrow U1$). Due to reconstruction error from concept decomposition, SMCS may not always be one, but is generally quite high (Fig. A8). We conduct the replacement test replacing the self-predicted coefficients and the cross-predicted coefficients. $\Delta$Pearson is the difference between the two similarity scores (SMCS and CMCS) and $\Delta$ L2, $\Delta$ KL and  $\Delta$ Match Acc are the differences between the outputs of the replacement tests for each. This is currently briefly described on Lines 319-320 and Line 421, but we will add a new clearer description in the final revision.

---

> > > > > ### Comment · Reviewer_73oU · 2024-11-29
> > > > >
> > > > > So as $\Delta$Pearson becomes closer to zero, the cross-model concepts are more similar to same-model concepts. This implies that the L2-distance (and KL should reduce) when $\Delta$Pearson becomes closer to zero. However, this trend is not esadily deducible from Fig.4. Can the authors please add more clarification?
> > > > >
> > > > > (I misread the legends in my intiial review)

---

> > > > > > ### Author Response · Authors · 2024-11-29
> > > > > >
> > > > > > The relationship between $\Delta$ Pearson and the output values (L2, Kl-div and Match Acc) is impacted by the concept's importance (measured using concept integrated gradients Sec. B.1.1).
> > > > > >
> > > > > > The same $\Delta$ Pearson can cause a larger change in L2/KL-div depending on how much importance the model places on the concept. We visualize importance using heat (brighter is more important). A small difference on an important concept (yellow) matters less than the same difference on an unimportant concept (black). For example, in Fig 4. $\Delta$ KL-divergence (middle column) we can see that changes on unimportant concepts make no impact on the logits.
> > > > > >
> > > > > > In addition, there is a possibility that concepts with the same importance and $\Delta$ Pearson have some differences in $\Delta$ L2 and $\Delta$ KL-div. This is because concept integrated gradients are measured for each sample in $I^c$, but the mean value is used as the concept's overall importance. Thus, if the concept $u^i_1$ and it's replacement $u^i_{2 \rightarrow 1}$ differ on images where $u^i_1$ is not important, there would be a smaller impact than if the differences occurred on important images. We discuss this point briefly from Line 418-420. In Fig. 4, we observe that within similar importance values and $\Delta$ Pearson values the effect on $\Delta$ L2 or $\Delta$ KL-Div tends to be similar. It is more clearly visible in the middle column.
> > > > > >
> > > > > > We will update the text to be clearer about this point and, as you previously suggested, we will add a plot to the appendix that separates the points by importance to better show the effect in the final revision.

---

### Official Review · Reviewer_fUV9 · 2024-10-31

**Soundness:** 3
**Presentation:** 4
**Contribution:** 3
**Rating:** 8
**Confidence:** 3

**Summary:**

The paper presents a method for comparing the representations of two neural networks using visual concepts. Concepts are extracted by factorising the matrix of activations corresponding to a set of images. The authors present two methods for the comparison of visual concepts, one based on correlations between the coefficient matrices, and another using regression, using one set of concept coefficients to predict the other and vice-versa. Also proposed is a method for measuring the dissimilarity of representations, where one network can contain a concept deemed important to classification, which is not present in the other network. These methods are evaluated on a variety of networks, investigating the relationship between concept importance and concept similarity.

**Strengths:**

- The topic is of great interest. Providing interpretable model comparisons that go beyond model performance will be useful to practitioners.
- The method goes beyond just measuring similarity, but provides interpretation for both similar and dissimilar concepts, something that is often overlooked in model comparison.
- The example in Figure 2 is very convincing.
- The proposed method for measuring similarity and dissimilarity is very intuitive.

**Weaknesses:**

- Some implementation details are missing (clarified in the questions section)
- Brainscore does link representation similarity scores to functional differences; models are evaluated using similarity in representations but also comparisons in the output. In places it is described as comparing representations between two neural networks, which is not true. It is between the activations in a network and  macaque brain activations.
- Colour bars in Figure 3 and 4 would make the figure slightly clearer.
- In Eq. 5, mentioning why L1 regularisation is used would make it clear to the reader.
- The introduction is quite brief, and fails to motivate the problem addressed. It could benefit from a small discussion on why interpretability is important and why concepts are more useful than say saliency maps or superpixels (LIME).

**Questions:**

-  How many images are used to construct  $\mathcal{I}_{1,2}^c$, and is it a requirement for the models to overlap slightly so that $\mathcal{I}^c$ has enough samples? How consistent are the concepts extracted with less samples? The images used to measure similarity seem very important for the ability to interpret results, as the images need to represent the concepts important for classification.
- Does the size of the patch effect the concepts found? And what patch size was used in the paper.
- Along the same lines, what is the split of $\mathcal{I}^c$ between test and train for the regression task?
- Do the authors have any intuition for using this method to compare models that do not necessary predict the same number of classes?
- In Figure 2, could you look at the regression weights learned, and see that the concept for the RN50 would be predicted by two separate concepts in the RN18? This would substantiate the claim in line 465, that possible the RN18 contains separate concepts for hands and net, as these concepts would have large contribution when predicting the 'hands/arms near a volleyball/net' concept in RN50.

---

> ### Author Response · Authors · 2024-11-23
> **fUV9 Response**
>
> Thank you for the review.
>
> > **[fUV9-1]** Missing implementation details.
>
> We provide further details about the approach in the revised Appendix B. We respond to specific questions below.
>
> > **[fUV9-2]** Interpretation of BrainScore.
>
> We see how our text may be considered misleading. Our aim is to highlight the uniqueness of the replacement test, which allows us to systematically test the impact of replacing columns of M2's coefficient matrix, with predicted coefficients from a regression model trained on M1's activation matrix. This test allows us to understand how conceptual differences in model representations lead to functional changes in model behavior. This kind of systematic variation is not conducted as one of the metrics in the BrainScore. We have modified the text in the introduction (first paragraph) to avoid making an incorrect claim about prior work.
> Yes, brain score compares ANNs to BioNNs, and we used the term NN as a general term to catch all types of neural network based computations. We add text to clarify this point (line 272).
>
> > **[fUV9-3]** Add colorbars to Figure 3 and 4.
>
>  We add colorbars to both figures.
>
>
> > **[fUV9-4]** Mention why L1 is used.
>
> We clarify the motivation for L1 regularization in equation 5.
>
> > **[fUV9-5]** Introduction is brief, please expand.
>
> We have expanded on the introduction to emphasize the motivation for our work, i.e. the potential impact of our method on explainable AI. Also, a discussion on local (saliency maps) and global (concepts) is provided in Section 2.2
>
> > **[fUV9-6]**  Better explain the image sets and patch size details.
>
> The number of images used to construct $I^c$ varies depending on the class and dataset. We enforce that there must be at least five images and at most we sample 200 images for concept comparisons. We use a patch size of 64x64 pixels resulting in 16 patches per image. Overlap is not required, since we take the union over the prediction sets from each model. You are correct that number of images, image grouping, and patch size would impact the concepts that are discovered. Future work should scrutinize the selection of concept proposal images further.  However, in this work we use an established protocol for concept generation (CRAFT) since our focus is on developing a tool for comparing concepts.  We include these details in Section B.1 of the Appendix.
>
> > **[fUV9-7]** What are the train and test splits for regression?
>
> For training the regression model, we use the train dataset of the deep neural network. Images in $I^c$ are split into five folds and five regression models are trained on each combination of four folds. For the final evaluation, the weights of all five regression models are averaged and the regression model is tested on images from the test set of the deep neural network. These details are available in Section B.2.
>
> > **[fUV9-8]** Intuition on grouping when models are not trained on the same set of classes.
>
> In this work, we choose to group images according to the model's predicted class. This choice is made because we are interested in features that contribute when the model believes an image belongs to a particular class. However, there are other reasonable options. For example, images could be grouped using unsupervised clustering in the latent space of the model.
>
>
> > **[fUV9-9]**  Exploration of regression weights.
>
> This is an interesting question, we have not yet explored this in this version of the manuscript. We will provide an update if we are able to complete this exploration. In addition, other reviewers found the volleyball class in Fig 2 to have a complex interpretation, so we have swapped it with a simpler example from the appendix that demonstrates a concept difference on the barbell class. We also have added several new experiments and generated concept difference visualizations for these experiments. Please see **[Wky2-1]** for details.

---

> > ### Comment · Reviewer_fUV9 · 2024-11-25
> > **Response from fUV9**
> >
> > Thank you for your changes. The introduction has been improved and many of the comments addressed. Appendix A.2 is a very nice addition and ideally would be included in the main text, however I understand the space constraints.
> >
> > >  However, in this work we use an established protocol for concept generation (CRAFT) since our focus is on developing a tool for comparing concepts.
> >
> > My concern was that CRAFT aims at explaining the concepts of one network, whereas the proposed method proposes comparing concepts between two networks, meaning that the construction of $I^c$ becomes important as it needs to be valid across both models in order for a valid comparison.

---

> > > ### Author Response · Authors · 2024-11-27
> > > **fUV9 Response Part 2**
> > >
> > > Thank you for your response. We have made one further update.
> > >
> > > > **[fUV9-9]**  Exploration of regression weights.
> > >
> > > In the new Fig. A14, we can see that the RN50 contains a concept for the volleyball class that has low similarity with the RN18. As suggested, we explored the weights of the regression model trained on the RN18 activations to see which neurons were used to try and reproduce the RN50 concept. We compute the permutation feature importance for the neurons to identify the top 6 most important neurons. We then visualize the image patches that most activated these neurons (Fig. A14). We find that neurons are sensitive to features like the boundary between volleyball nets and background, volleyballs in the sky, players near volleyball nets with a focus on lower bodies, and close-up pictures of nets with hands and arms. We also note that the most important neuron seems to activate for objects that appear to be **like** the boundary between a volleyball net and the background, but are in reality not a volleyball net at all. For example, the neurons activate strongly for a grid texture on a wall (see Neuron 465). When compared to the over-predicted images, the contributing neurons seem to qualitatively explain the increased sensitivity to certain types of images.
> > >
> > > > **[fuV9-10]** Construction of $I^c$
> > >
> > > We agree that this is an important point. We will update the final text to emphasize the importance of constructing a valid image set. Thanks for the pointer!

---

### Official Review · Reviewer_RkXK · 2024-11-02

**Soundness:** 2
**Presentation:** 3
**Contribution:** 3
**Rating:** 6
**Confidence:** 4

**Summary:**

The paper presents an approach for measuring the similarity between representations of two models using "concepts". It quantifies the similarity between the internal concepts of the models, extracted with a concept-based xAI method.

**Strengths:**

1. The paper is well-written, easy to follow, and defines well the different stages of the method and the evaluation.

2. The idea of the paper is novel - instead of comparing functional similarity or direct similarity in representations, it uses a decomposition into "concepts" and compares the similarity between concepts. Instead of providing one similarity measure, this paper also demonstrates visually what are the concepts that make two models similar or dissimilar.

3. The paper provides a comprehensive study of the relations between concept similarity and concept importance to address the main possible limitation of using concepts - their non-straight-forward connection to the computation in the model.

**Weaknesses:**

1. This approach might be sensitive to the concept extraction method. The extraction of $U_i$ and $W_i$ provide very different results for different concept dictionary sizes $k$ and if some L1 sparsity regularization is applied. A comparison of different approaches for concept extraction and ablation of their hyper-parameters can provide better insights into the approach sensitivity.

2. Computational cost - this process requires computing representations on many images, extracting representations from each layer, training linear models, and comparing correlations. The computational cost of this process is not discussed, and the trade-offs between the number of images that are needed for each of the steps are also missing.

3. Usability - The current approach for coming up with meaningful observations about the representations requires a human in the loop, who goes over all the concepts and investigates the similar and dissimilar concepts (as shown in Figure 2). Can it be automated somehow?

4. Missing related work:

Dravid et al., Rosetta neurons: Mining the common units in a model zoo, ICCV 2023

Rombach et al., Network-to-network translation with conditional invertible neural networks, NeurIPS 2020

Typo in 250: We use it *to* compute...

**Questions:**

Why does the approach use per-class concepts? Different classes of images can have shared computation and shared intermediate concepts? Using per-class concepts might result in missing shared concepts, and functional similarity.

The current model assumes implicitly that the models $M_1$ and $M_2$ are classifiers (the matching is applied per class). What modifications are needed to make it work on other types of models (e.g. representation learners/segmenters)?

What data is used to compute the linear mapping in line 282? Is it part of $I^c$?

---

> ### Author Response · Authors · 2024-11-23
> **RkXK Response**
>
> Thank you for your review.
>
> > **[RkXK-1]** Sensitivity to concept extraction method.
>
> We agree that the approach may be sensitive to the concept extraction method. In this work, we used a concept extraction approach that had been previously tested in CRAFT (Fel et al. CVPR 2023) and focused on developing a tool to compare the concepts. In Fel et al. NeurIPS 2023 , the authors show that NMF has desirable trade-offs for several properties: reconstruction error, sparsity, stability, FID, and OOD generalization. We have conducted an ablation of the L1 penalty in the appendix. We find that the histogram of similarity values is consistent for lamba values 0.01, 0.05, and 0.1, but that similarity values drop significantly when going to 0.5. In our work, we choose 0.1.
>
> > **[RkXK-2]**  Computational cost.
>
> We include a breakdown in Table A3 in the Appendix (Sec. B) detailing the computational cost of each step of our approach. Overall, comparing a ResNet-18 to a ResNet-50 using both RSVC and MMCS takes around 1.2 minutes per class and 20 hours for all 1000 classes of ImageNet on a single machine with a 4090 GPU. We also include details about the number of images used in each step of the approach in Section B.
>
> > **[RkXK-3]**  Usability and automation of concept difference interpretation.
>
> We include a new experiment in Appendix A5 using ChatGPT-4o to interpret the image collages generated by our approach. We find that the interpretations provided by the LLVM are similar to the manual annotations we have provided previously.
>
> > **[RkXK-4]**  Missing related work and typo.
>
> Rosetta Neurons (Dravid et. al 2023) are most similar in spirit to our work. This method correlates the activation patterns of neurons in two different networks. They seek out neurons with similar activation patterns and coin these neurons to be “Rosetta Neurons”. However, they do not aim to provide a method to test for dissimilar neurons that significantly contribute to model behavior, which is the primary focus of our work. In addition, Rosetta Neurons are found through correlation, which can miss many-to-one mappings from one model to the other. Network-to-Network translation (Rombach et. al 2020) uses an invertible neural network to map between the latent space of a language model and a generative vision model. Their approach focuses on building a tool for text-to-image generation. Our work targets a different problem (explainability) and is designed to accurately quantify and visualize the difference between two models. Thank you for pointing out these references, we include them in the updated manuscript and discuss the differences between our method and Rosetta Neurons (Line 111). We have corrected the typo as well.
>
> > **[RkXK-5]** Why does the approach use per-class concepts?
>
> We use per-class concepts because it helps organize and simplify concepts for the user, however, other grouping strategies are valid and interesting directions. As you have pointed out, grouping across the dataset can help reveal shared computations. In the future, we would like to explore grouping commonly confused classes to discover more complex model behavior.
>
> > **[RkXK-6]** The current approach assumes the models are classifiers.
>
> For non-classification problems users would have to select a method to group images. A simple option in a setting where there are no class labels is to perform unsupervised clustering in the latent space to find clusters of semantically related images. Alternatively, one can use a probe classification dataset to compare SSL representations, as we do for the DINO and MAE models in Figures 3 and 5.
>
>
> > **[RkXK-7]** What data is used to compute the linear mapping in line 282?
>
> Images for the linear regression are sampled from $I^c$. We provide further implementation details in Section B.2.

---

> > ### Author Response · Authors · 2024-11-27
> > **RkXK Response Part 2**
> >
> > > **[RkXK-1]** Sensitivity to concept extraction method.
> >
> > We have improved the parameter sweeps. First, we visualize the impact of the L1 penalty on the similarity scores (Fig. A12). We see that, as expected, increasing the L1 penalty increases the number of zero coefficients (middle column) and lower similarity scores (left column). In this main paper, we have used an L1 penalty of 0.1. Second, we visualize the impact of different numbers of concepts (Fig. A13). We see that increasing the number of concepts (e.g., >10) reduces the reconstruction error (middle column), but also results in more low/medium similarity concepts (left column). We use 10 concepts for experiments in the main paper.

---

> > > ### Author Response · Authors · 2024-11-30
> > >
> > > Greetings RkXK,
> > >
> > > Do not hesitate to let us know if you have any additional questions regarding our response to your review.

---

### Official Review · Reviewer_wky2 · 2024-11-03

**Soundness:** 2
**Presentation:** 4
**Contribution:** 3
**Rating:** 8
**Confidence:** 4

**Summary:**

This paper introduces Representational Similarity via Visual Concepts (RSVC), a method to compare representations between neural networks by analyzing shared and unique visual concepts instead in pairs of pretrained models, with the goal of providing a better comparative breakdown of pairwise model similarities.
Conceptually, RSVC extends work by Fel et al. 2023a, b on a dictionary-based formulation of concept-based XAI, and introduces different mechanisms to study similarities between concept explanations of two different networks: (1) A correlational analysis of concept coefficients, and a (2) regression-based approach, in which a linear model is trained to predict concept assignments between models. These two concept similarity methods are then used in conjunction with a concept importance estimator and a concept-performance-dependence study to broadly understand the existence and prevalence of unique and shared concept pairs between models.
Experiments are conducted for RN18, RN50, ViT-S, ViT-L, MAE & DINO pretrained on ImageNet to showcase concept similarities between different model architectures and training paradigms.

**Strengths:**

In general, I find this paper to be sufficiently well written and structured; with the presentation and detailed image visualization and captions to be commendable: Figures and their captions can be read standalone and still convey key contributions; which significantly enhances readability. Moreover, the proposed method is described and visualized in a way that makes it much easier to grasp all relevant key aspects.

At the same time, the extension of works by Fel et al. seems sensible, and to the best of my knowledge, the use of a dictionary-based concept attribution method for __model similarity__ studies is novel.

Moreover, I very much appreciate the experimental breadth the authors followed to test the different potential usecases of RSVC.

**Weaknesses:**

Unfortunately, I currently have some more pressing issues and questions which stop me from recommending acceptance at this stage. Ordered by importance, these are:

---

__Lack of sanity checks and proof of concepts__.
Interpretability methods are strongly reliant on sanity checks (Adebayo et al. 2018, https://arxiv.org/abs/1810.03292, Boehle et al. 2022, https://arxiv.org/abs/2205.10268). The authors unfortunately directly apply their proposed methods on arbitrary model pairings, without first firmly establishing the validity of the proposed approach. To do so, I believe that at the very least, the following steps need to be included:

(1) Establishing a baseline. This means that using two models for which concept similarities are highly likely or even guaranteed, and applying both MMCS and RSVC (regression-based) to understand how visualizations in Fig. 3 changes. As it is now, it is much more difficult to relate computed values and their respective significance. A baseline could e.g include the same model architecture trained on the same data ordering, but with small changes in the initialization; or a fixed initialization but changes in the data ordering.
(2) Studying variations away from the baseline above. By structurally varying the baseline (e.g. going to different augmentations, different data subsets, ...) understand how the proposed metric changes.
(3) More clearly establish what extracted concepts actually look like. The example in Fig. 1 with bird tail and sky background seems quite convincing, but then when looking at actual examples in Fig. 2, these become much less convincing: The net appears consistently in all scenarios (alongside arms and hands), and I find it difficult to see any consistency here.

Without establishing such baselines and sanity checks, it is very difficult to assign actual meanings to extract values.

---

__Conceptual / Architectural Inconsistencies__.
There are some elements in the proposed approach that currently do not make much sense to me, particularly:

* L171: "Each concept proposal is resized to the model’s input resolution and passed through the network." >>> Learned features are generally resolution-dependent unless speficially trained not to be. As such, the resulting extract concepts (from the resized image patches) hold much less meaning with respect to the actual network, no? I may not understand the setup sufficiently here, but resizing patches and using thereby derived U_i, W_i for concept comparisons does not make much sense to me. Why not just store the full activation output and break it down corresponding to the patch formulation?


* As the authors correctly state for the correlational setup described in 3.2.1, MCS does not accout for any confounding elements. But it is not completely clear to me how RSVC avoid the issue of confounders? If I have a confounding element in e.g. M2 which would produce similar concept coefficient than the actual extracted concept would, this would still be an issue even when using the learned linear map, right?

* For the replacement tests: What are the guarantees that U_2-1 operates in the same basis as U_1, such that a direct replacement is well defined? The learned maps in Eq. 4 are trained without any guarantees that the bases between both models overlap, no?

---

__Some less important notes that are not majorly impacting my decision, but are useful to address:__

* L133-135: "While our work also aims to compare two models in an interpretable manner, our approach does not make use of surrogate models and instead uses the activations of the original model itself." But isn't the projection matrix another learned linear model? I.e. isn't RSVC also using surrogate models?

* The use of a linear predictor (or one with directly assignable importance values) to test matching up to permutation, scale and shift has been studied in-depth in disentangled representation learning, e.g. in Locatello et al. 2018 (https://arxiv.org/abs/1811.12359), Eastwood et al. 2018 (https://openreview.net/forum?id=By-7dz-AZ), Locatello et al. 2019 (https://jmlr.org/papers/volume21/19-976/19-976.pdf), Eastwood et al. 2022 (https://arxiv.org/abs/2210.00364) or Roth et al. 2023 (https://openreview.net/pdf/f1f46edad3b1c36e9c46c00c16e8592cfd1e4df6.pdf). These may provide a useful resource for experimental ablation studies, and should be discussed to augment the short reference to disentangled representation learning in the related works section.

* Section 3.3 is fairly redundant, and should either be excluded entirely, or in my eyes more crucially, described in much more detail, as the use of concept importance is quite prevalent throughout the paper.

**Questions:**

See weaknesses. I'm happy to raise my score if the authors can address my first two points, and during the rebuttal can provide baseline studies and sanity checks; as I believe the paper to be an overall meaningful contribution if these points can be incorporated.

---

> ### Author Response · Authors · 2024-11-23
> **Wky2 Reponse Part 1**
>
> Thank you for the review.
>
> > **[Wky2-1]** “Lack of sanity checks and proof of concepts”.
>
> We agree that establishing the validity of the approach is very important. Following the reviewers’ suggestion, we conduct six new experiments that we detail below. These experiments can be found in the appendix and are marked in blue.
>
> (1) Toy Concept Experiment (Sec. A.2, Figs. A.2, A.3) : We train two models from scratch on modified versions of the NABirds dataset. Model 1 (M1) is trained on a dataset in which a pink square is randomly placed on images of the Common Eider class with a 70% frequency. Model 2 (M2) is trained with a pink square appearing 50% of the time on images of any class, giving the pink square no predictive ability. Given this training formulation, we expect M1 to react to pink squares and M2 to ignore them. We apply RSVC to these models and show that (1) we discover a pink square concept in M1 that has a ~0 similarity score to M2 and (2) that other concepts for the Common Eider class are preserved and have higher similarity.
>
> Seed & Dataset Variation (Secs. A.3, A.4, Figs. A4, A5, A7) - We train a ResNet18 model from scratch on the NABirds dataset. The features of this model are used as a baseline for comparisons as other aspects of training are varied. The experiments are ordered by increasing concept dissimilarity.
>
> (2) (Fig. A4) We train and compare to another ResNet18 using a different seed (same initialization).  We find that the similarity between the two models is still quite high.
>
> (3) (Fig. A4) We train a ResNet18 on a modified version of NABirds in which the waterbirds (169/555 classes) are excluded.  We find that more dissimilar concepts arise, but the overall similarity is still high.
>
> (4) (Fig. A4) We train a ResNet18 on a combined dataset of NABirds and Stanford Cars. We find that the introduction of qualitatively different data, results in learning significantly more dissimilar concepts. We conduct a visual analysis of the features learned from these models. (Fig. A7).  We find intriguing differences in the types of features learned. The model trained on both datasets contains complex car features that are specific to a car model. For example, we find a feature that activates for the racing stripe of a specific muscle car. In contrast, we find that the other model uses simpler features that activate for specific colors and car styles.
>
> (5) (Fig. A4) We compare to a model pre-trained on ImageNet and fine-tuned on NABirds.  We find that, expectedly, ImageNet introduces a significant amount of dissimilar concepts.
>
> (6) (Fig. A5) We also explore the effects of varying seed when finetuning DINO and MAE models on NABirds. We find that pre-training method has a larger impact on similarity than seed.
>
> We found these results to be intuitive, with conceptual similarity decreasing with larger modifications of the training protocol.
>
> > **[Wky2-2.1]** Do patches make sense?  Why not patch the activations after passing the image through the network?
>
> All models are trained with random resized crop or inception style cropping with the smallest crop being smaller than the patch size. This makes the patches in-domain for the network. We clarify this point in the text. It is an interesting idea to patch the activations after passing an image through the network. As far as I am aware, this approach has not been tried. In this work, we made use of an established protocol for concept extraction (CRAFT) that had been tested with human experiments, but other approaches could be effective.
>
> > **[Wky2-2.2]** How does RSVC avoid confounders?
>
> There are two ways that confounders can be introduced: (1) Through visual features that co-occur in the images. For example, a beak and an eye will appear together in most images of birds or (2) the concept extraction method entangles two visual features even if the network encodes them independently. To address (1), we make use of image patching which helps us break down correlated visual features. This does not guarantee that all visual features become uncorrelated. As suggested in the review, both RSVC and MCS would be impacted by this type of confounder. In the paper, we highlight that MMCS fails with respect to (2), the concept extraction process can lump together neurons that are actually encoding different things. L262: "However, correlation based similarity can be affected by confounds from concept extraction". In contrast, RSVC uses lasso regression directly from the activation matrix A1 and avoids the issue of confounds introduced by the decomposition of M1.

---

> ### Author Response · Authors · 2024-11-23
> **Wky2 Response Part 2**
>
> > **[Wky2-2.3]** Replacement test basis?
>
> RSVC learns a map from A2 to U1. This follows the intuition that, if a concept from M1 is encoded by M2, there should be a neuron or group of neurons in M2 that can recreate the concept coefficients from U1. The concept basis for M2 does not play a role in this mapping. Please do not hesitate to respond if we have not interpreted the question correctly.
>
> > **[Wky2-3.1]** Is RSVC also a surrogate model?
>
> Thank you for pointing this out. After consideration, we believe that you are correct that technically RSVC also uses a surrogate model due to the NMF decomposition. We have reworded the sentence to remove the word surrogate, but still emphasize the critical advantage of our method, which is using the network's activations directly.
>
> > **[Wky2-3.2]** Linear predictors have been used in the disentangled representation learning literature.
>
> Thank you for highlighting these methods, we have added them to the related works section.
>
> > **[Wky2-3.3]** Section 3.3 is fairly redundant.
>
> We move the details of Section 3.3 to the appendix for space considerations.

---

> > ### Author Response · Authors · 2024-11-27
> > **Wky2 Response Part 3**
> >
> > **[Wky2-1.2]** Fig. 2 contains a challenging visual concept to interpret.
> >
> > Concepts vary in their complexity, so we have swapped Fig. 2 for a clearer example using a concept from the “Rugby” class. To understand the more “complex” volleyball concept, we include a deeper analysis (Fig. A14) in this new revision. The RN50 contains a concept for the volleyball class that has low similarity with the RN18. We explore the weights of the regression model trained on the RN18 activations to see which neurons were used to try and reproduce the RN50 concept. We compute the permutation feature importance for the neurons to identify the top 6 most important neurons. We then visualize the image patches that most activated these neurons. We find that neurons are sensitive to features like the boundary between volleyball nets and background, volleyballs in the sky, players near volleyball nets with a focus on lower bodies, and close-up pictures of nets with hands and arms. We also note that the most important neuron seems to activate for objects that appear to be **like** the boundary between a volleyball net and the background, but are in reality not a volleyball net at all. For example, the neuron activates strongly for a grid texture on a wall. When compared to the over-predicted images, the contributing neurons seem to qualitatively explain the increased sensitivity to certain types of images.
> >
> > **[Wky2-1.3]** More clearly establish what extracted concepts actually look like.
> >
> > The revisions contain more examples of concepts. We are happy to include more in the final revision if desired.

---

> > > ### Author Response · Authors · 2024-11-30
> > >
> > > Greetings Wky2,
> > >
> > > Do not hesitate to let us know if you have any additional questions regarding our response to your review.

---

> > > > ### Comment · Reviewer_wky2 · 2024-12-02
> > > > **Response to Rebuttal**
> > > >
> > > > I thank the authors for their thorough rebuttal, which has significantly increased the quality of the contribution, and has answered most of my raised concerns:
> > > >
> > > > * The authors have included a *significant* array of additional sanity checks, which have notably helped support the validity of RSVC,
> > > > * have included additional examples for extracted concepts,
> > > > * better discussed Fig. 2,
> > > > * and correctly placed RSVC as a method reliant on a surrogate model (though providing sufficiently convincing argumentation for RSVC).
> > > >
> > > > Having also gone over all the other reviews and their respective rebuttals, I believe that the paper has now I reached a stage where I am happy to advocate for acceptance. Consequently, I have raised my result from 5 to 8 to indicate this change.

---

### Official Review · Reviewer_8D1Q · 2024-11-05

**Soundness:** 3
**Presentation:** 3
**Contribution:** 2
**Rating:** 6
**Confidence:** 3

**Summary:**

This paper studies the representational similarity between different vision networks. The main novelty is to provide insight into not just _how_ similar are two networks but in _what_ ways are they similar. This is achieved through a dictionary learning method that approximates the activations at some layer of a network as a nonnegative combination of "visual concepts", which are just a set of activation vectors.  Then the similarity between two networks is measured as the similarity of the visual concepts they each represent, either through a correlation-based metric or via regression from one set of concepts onto the other. The paper finds that there is some degree of similarity between the visual concepts in different networks, that there are some important concepts that differ between different networks, and that layers near the input or near the output of the network are most alike, while middle layers differ.

**Strengths:**

I like the motivation of this paper. It is convincing that 1) understanding _what_ differs between different neural representations is important,  and 2) prior work has mostly overlooked this question (although a few papers that were not cited do address this question; see weaknesses).

Other strengths include:
* The writing is mostly clear.
* The methods appear to be reasonable and I don't see obvious errors.
* The paper delivers a tool that could increase interpretability of representational differences, because the differences can be visualized.
* Several of the findings are interesting; I was especially intrigued by 1) the method can identify low similarity, high importance concepts, 2) middle layers show the greatest differences between different networks.

**Weaknesses:**

The main premise of this paper is to go beyond prior work on representational similarity, which only gave summary numerical scores, and instead show "_what_ visual concepts make two models similar or dissimilar." This is a great question, but the paper falls short of fully delivering on it.

There is only one figure, Fig 2, in the main paper that directly addresses this central question, comparing two models and showing how they differ in terms of the visual concepts they learn. The appendix has more results along these lines but I'm still not sure what insights to take away from these results.

I take it to be that the contribution is not to *show* what is different between different nets but rather to provide a *tool* that is in principle capable of showing what is different. This is a valid contribution but I think the paper would nonetheless be stronger if the tool were used to provide new insights into the model differences.

The other results (Figs 3, 4, and 5) are interesting but fall back into the category of summary statistics that don't go deep into what makes two models similar or dissimilar. Instead we only learn that 1) there are important differences (intriguing! but what are they), and 2) the differences live primarily on the middle layers of networks (although from Fig 5 it looks like the increase in similarity at the deepest layers is quite minor indeed).

If this paper is indeed meant to just introduce a tool, rather than a discovery, then the reader needs to be convinced that the tool is valuable. I'm mostly convinced that the tool is valid and in some ways new. But I'm not sure what to do with it, nor if the community would be able to use this tool to make new discoveries. This isn't an absolutely critical weakness -- time can tell if it is useful -- but I would rather have seen more validation of the usefulness of the tool in this paper. It would be wonderful if the paper could use the tool to tell us something about neural representations we didn't know before, like, say, that resnets focus more on background elements than transformers, or, ideally, something deeper than that. To make this super compelling, I would like a demonstration that this new way of identifying differences has advantages over other ways of identifying representational differences, in a head to head comparison. But I'm not sure how to do that.

There are also a few overlooked prior works I think are worth discussing (or, ideally, comparing to):

1. "Rosetta Neurons: Mining the Common Units in a Model Zoo," Amil Dravid, Yossi Gandelsman, Alexei A. Efros, Assaf Shocher, ICCV 2023

-- This paper also identifies visual concepts that are shared between different models. In their case, the concepts are coded by neurons, across different models, that are correlated in terms of their responses to the same set of image patches.

2. "Understanding the Role of Individual Units in a Deep Neural Network," David Bau, Jun-Yan Zhu, Hendrik Strobelt, Agata Lapedrizad, Bolei Zhouf, and Antonio Torralba, PNAS 2020

-- This paper also provides a tool for identifying which visual concepts are encoded by different networks, and compares two networks in this way.

3. "Exploring Neural Networks with Activation Atlases," Shan Carter, Zan Armstrong, Ludwig Schubert, Ian Johnson, Chris Olah, Distill, 2019

-- This paper introduces "activation atlases", which are yet another way of identifying visual concepts learned by different networks. This work does not use these atlases to compare different models, which differs from the current paper's goals.

There is also a recent line of work on interpretability via sparse autoencoders that may be worth looking into, e.g., https://transformer-circuits.pub/2024/scaling-monosemanticity/

**Questions:**

1. I didn't fully understand why use image patches rather than whole images. This could be motivated more clearly, or justified with an experiment comparing using patches vs whole images.
2. I would appreciate if the authors can address the missing reference I mentioned above. Why should a reader prefer your proposed method over these prior methods? What are the advantages and disadvantages of each approach, or are they not comparable?

---

> ### Author Response · Authors · 2024-11-23
> **8D1Q Response Part 1**
>
> Thank you for your review.
>
> > **(8D1Q-1)** What is the core goal of this work? Is it to present a tool?
>
> We agree that *one* of the main contributions of this work is to present a tool for surfacing model differences. In this work, we contribute by (1) defining the problem of visualizing model differences, (2) providing a tool to solve the problem, and (3) establishing that the tool measures what we intend it to measure.
>
> > **(8D1Q-2)** Novelty and usefulness of the tool.
>
> We have updated the manuscript to clarify the importance of this problem for the field of XAI. In general, XAI methods sacrifice model fidelity to produce explanations that are simple enough for human interpretation. While simplification is important, it’s not clear how much is appropriate. For example, it is possible that simplification leads to indistinguishable explanations for all models trained on the same data. On the other hand, preserving complexity leads to a large number of neurons or concepts to inspect, making it extremely challenging to understand why a model makes a particular decision and which features are actually interesting. Our work is **novel** as it is the first to take complex models, without any specific training to disentangle the representations, and show that XAI methods can be adapted to find features that are *unique* and *important* to that model. Our approach is **useful** as it allows users to filter complex explanations for interesting features that contribute meaningfully to behavior. We hope that, in the future, this can be used to give complete explanations for model failures, an unsolved challenge in XAI [R1].
>
> [R1] Colin et al., I Do Not Understand: A Human-Centered Evaluation Framework for Explainability Methods, NeurIPS 2022
>
> > **(8D1Q-3)** Value of Figures 3, 4, and 5.
>
> We are pleased that you find insights from Figs 3, 4 and 5 interesting. We include these figures because they answer critical questions that assess the validity and usefulness of the proposed tool. Fig 3 shows that RSVC can discover concepts that are *unique* and *important* to a model, Fig 4 links representational differences in two model’s to specific changes in a model’s behavior, and Fig 5 (1) verifies existing observations in the literature (2) provides a different perspective into how layers relate in terms of concepts. As a small step in trying to gain deeper insights, we provide a new experiment in the Appendix A3 comparing a model trained on NABirds to a model trained on NABirds *and* Stanford Cars. We find that there are some consistent differences in the types of features learned. The model trained on both NABirds and Stanford Cars exhibits complex concepts that are semantically aligned with specific car models. For example, one discovered concept is sensitive to the racing stripe on a type of muscle car. In contrast, car concepts for the model trained solely on NABirds appear to be sensitive to a particular color and style of car, but are not specific to the car model. RSVC opens the door to future focused evaluation on a specific model and a small set of classes to present a complete picture of model behavior. In this work we have focused on a broader exploration of the tool's properties to demonstrate its potential.

---

> ### Author Response · Authors · 2024-11-23
> **8D1Q Response Part 2**
>
> > **(8D1Q-3)** Discussion of additional prior works.
>
> We included the suggested references in the latest version of the manuscript (Sec 2). Bau et. al PNAS 2020 is designed for single networks, to the best of our understanding. The authors visualize the activations of a convolutional neural network and use an object detector to decide how well the activations match a semantic concept. In order to compare two networks, the authors compare the statistics of the detected concepts, but this is limited by the quality of the detector. In addition, detectors are trained on datasets of known concepts, but networks can discover and use novel and abstract concepts. The work of Dravid et. al ICCV 2023  is most similar in spirit to our work.  Their work correlates the activation patterns of neurons in two different networks. They seek out neurons with similar activation patterns and coin these neurons to be “Rosetta Neurons”. However, they do not aim to provide a method to test for dissimilar neurons that significantly contribute to model behavior, which is the primary focus of our work. In addition, Rosetta Neurons are found through correlation, which can miss many-to-one mappings from one model to the other. Finally, activation atlases (Carter et. al Distill 2019), are an excellent tool to visualize model concepts as a whole. The focus of their method is to develop a 2-D map of feature visualizations that is organized in a way that reflects the representation of the network. Different from RSVC, their work is not focused on quantifying and precisely measuring the differences between two models, rather they are seeking a broader, more general picture of network behavior. It would be interesting to develop an atlas that could be used for understanding model differences in the future. Due to these key differences between our work and the prior works, we believe that our work is not comparable. Thanks also for sharing the sparse autoencoders references, since our method operates using concept dictionaries, sparse autoencoders could be used instead of NMF in future explorations.
>
>
> > **(8D1Q-4)** Why patches?
>
> We follow prior work (i.e. CRAFT) in using image patches because they help improve interpretability for the user. When using entire images, some visual features will nearly always co-occur. For example in the bluejay class, almost every image will contain the tail, body and beak. By splitting into patches we can identify that the model is specifically reacting to the tail. All models are trained with random resized crops, making the patches in-domain for the network. We have added a line to justify this choice in the text (Line 177-178).

---

> > ### Author Response · Authors · 2024-11-30
> >
> > Greetings 8D1Q,
> >
> > Do not hesitate to let us know if you have any additional questions regarding our response to your review.

---

> > > ### Comment · Reviewer_8D1Q · 2024-12-01
> > > **post-rebuttal update**
> > >
> > > Thanks for the clarifications and extensive new experiments. I'm raising my score to reflect the improvements.
> > >
> > > I think the paper is significantly improved. The scope and goal comes across more clearly in the new text. The experiments in the appendix help a lot on validating the meaningfulness of the method. My favorite new experiment is the toy experiment with the purple box. I would move this to the main text. That's the most convincing validation to me, since there is a "ground truth" you can compare to.
> > >
> > > The real data results remain hard for me to interpret. For example, in Fig 2, there hands and weight equipment in both the under- and over-predicted montages... it's not completely obvious how to intrepret this interpretation. My main concern remains that I'm not sure the tool highlights _important_ and _actionable_ differences between models. We get to see a long list of specific differences between the visual concepts two nets encode, but what am I to make of this, how will it push our understanding or engineering forward? Indeed, contrary to the motivation of this paper, what I might often prefer is a summary statistic about the main differences (like: "this model has a texture bias") rather than a detailed look at all these specific differences. To improve the paper further, I think it would help to show a real world use case, or a user study. That said, I'm okay with leaving these steps for time to tell.
> > >
> > > I like the use of ChatGPT to translate the visual differences into a text summary. This approach reminds me of the following paper, which might be referring to:
> > >
> > > "Interpreting CLIP's Image Representation via Text-Based Decomposition"
> > > Yossi Gandelsman, Alexei A. Efros and Jacob Steinhardt
> > > ICLR 2024

---

> > > > ### Author Response · Authors · 2024-12-01
> > > >
> > > > Thank you for looking at our revision. We are glad that you think the updated text and experiments improve the paper. We agree that the toy concept (purple square) experiment is striking, we will try to restructure the work and lift this experiment to the main text.
> > > >
> > > > We hope that future work will improve the tool and help make these insights more actionable.
> > > >
> > > > Thank you for the reference, we will include it in the final revision.

---

### Author Response · Authors · 2024-11-23
**General Response**

We thank the reviewers for their feedback which we found very useful. Below we recap reviewer feedback and summarize changes in the manuscript that address weaknesses. We also provide direct responses to individual reviewers.

We uploaded a revised version of the paper addressing the reviewer comments. In the revised manuscript, new additions are colored blue and changes are colored in red.

Reviewers found that our work was well-motivated and that the method presented was sensible.

They asked us to stress-test our method further and conduct more controlled experiments to reveal interesting features in the models. In the revised manuscript we provide seven new experiments in which we carefully vary the training protocol and measure model similarity. The new experiments include demonstrating that it is possible to automatically generate text descriptions of the discovered model differences (see Fig A6), a toy experiment that demonstrates that we can uncover clear differences in the representations learned by different models (see Fig A2, A3), and a clearer qualitative example (see Fig 3).

They asked us to improve the abstract, introduction, and related work. In the revised manuscript, we clarify the goals of the work and discuss additional important, but different, related works.

They asked for more implementation details. We add more information about how concept extraction and concept similarity are computed. We include a detailed breakdown of computational cost (Section B) and add an ablation experiment for Lasso Regression (Fig. A12).

---

> ### Author Response · Authors · 2024-11-27
>
> Thanks again to the reviewers for their feedback. We have added a final revision of the paper where we add some of the additional analysis requested in the original reviews. Specifically, we include the following which can be found in the revised PDF:
>
>
> - In response to **RkXK** and **73oU** request we add parameter sweeps over the L1 parameter and the number of concepts (Fig. A12, Fig. A13). The new results show that increasing L1 penalties results in more zero coefficients and lower similarity values. Increasing the number of concepts decreases the reconstruction error, but also results in more low similarity concepts.
>
> - **wky2** and **73oU** found that the volleyball concept was challenging to interpret. We have replaced the main figure with a clearer example. However, to better understand the volleyball concept, we follow **fUV9**’s suggestion and include a deeper analysis of the neurons in the RN18 used to predict the volleyball concept in RN50 (Fig. A14). This visualization uses permutation importance to identify the most important neurons for the regression model. The neurons demonstrate a sensitivity to a variety of visual features, including the boundaries between nets and the background, volleyballs in the air, players in gyms, and close-ups with nets with hands nearby. These neurons qualitatively seem to explain some of the over-predicted samples.

---

### Meta-Review · Area_Chair_jMhX · 2024-12-05

**Metareview:**

This paper works on the interpretability of deep neural networks. It introduces an interpretable representational similarity method (RSVC) to compare two networks. The proposed method discovers shared and unique visual concepts between two models. A wide range of neural networks are compared.

Strengths:
- Novel Approach: Introduces a dictionary-based method to compare neural network representations through visual concepts, offering a more interpretable way to analyze model similarities and differences.
- Clear Presentation: Well-structured paper with detailed visualizations, making the methodology and findings easy to follow.
- Interesting Findings: Highlights new insights, such as middle layers exhibiting the most representational differences and the identification of high-importance, low-similarity concepts.

Weaknesses
- Limited Validation: Lacks sufficient baseline studies, sanity checks, and comparisons with prior methods to establish the validity and practical advantages of the approach.
- Usability Concerns: Heavy reliance on manual investigation of concepts and absence of automation, making it challenging for broader community adoption.
- Incomplete Insights: Fails to provide strong new discoveries about model differences and overlooks related work, which limits the impact and clarity of the contribution.

Reviewers raised several issues, and the authors prepared a detailed rebuttal. Most of the issues have been addressed. There is a consensus of acceptance. Thus acceptance is recommended.

**Additional Comments On Reviewer Discussion:**

The majority of the concerns from the reviewers have been addressed. There is a consensus of acceptance from the reviewers.

---

### Decision · Program_Chairs · 2025-01-22

Accept (Poster)